# Endo-lysosomal Aβ concentration and pH trigger formation of Aβ oligomers that potently induce Tau missorting

Marie P. Schützmann[1,6], Filip Hasecke [1,6], Sarah Bachmann [2,6], Mara Zielinski [3], Sebastian Hänsch[4], Gunnar F. Schröder [3,5], Hans Zempel [2✉] & Wolfgang Hoyer [1,3✉]

Amyloid-β peptide (Aβ) forms metastable oligomers >50 kDa, termed AβOs, that are more effective than Aβ amyloid fibrils at triggering Alzheimer's disease-related processes such as synaptic dysfunction and Tau pathology, including Tau mislocalization. In neurons, Aβ accumulates in endo-lysosomal vesicles at low pH. Here, we show that the rate of AβO assembly is accelerated 8,000-fold upon pH reduction from extracellular to endo-lysosomal pH, at the expense of amyloid fibril formation. The pH-induced promotion of AβO formation and the high endo-lysosomal Aβ concentration together enable extensive AβO formation of Aβ42 under physiological conditions. Exploiting the enhanced AβO formation of the dimeric Aβ variant dimAβ we furthermore demonstrate targeting of AβOs to dendritic spines, potent induction of Tau missorting, a key factor in tauopathies, and impaired neuronal activity. The results suggest that the endosomal/lysosomal system is a major site for the assembly of pathomechanistically relevant AβOs.

[1] Institut für Physikalische Biologie, Heinrich-Heine-Universität Düsseldorf, Düsseldorf, Germany. [2] Institute of Human Genetics and Center for Molecular Medicine Cologne (CMMC), University of Cologne, Faculty of Medicine and University Hospital Cologne, Cologne, Germany. [3] Institute of Biological Information Processing (IBI-7) and JuStruct: Jülich Center for Structural Biology, Forschungszentrum Jülich, Jülich, Germany. [4] Department of Biology, Center for Advanced Imaging (CAi), Heinrich-Heine-Universität Düsseldorf, Düsseldorf, Germany. [5] Physics Department, Heinrich-Heine-Universität Düsseldorf, Düsseldorf, Germany. [6] These authors contributed equally: Marie P. Schützmann, Filip Hasecke, Sarah Bachmann. ✉email: hans.zempel@uk-koeln.de; wolfgang.hoyer@hhu.de

Aβ amyloid fibrils are highly stable protein aggregates of regular cross-β structure that constitute the main component of the senile plaques in the brains of Alzheimer's disease (AD) patients[1–3]. Although amyloid fibrils can exert toxic activities, metastable Aβ oligomers are thought to represent the main toxic species in AD[3–5]. At sufficiently high monomer concentration, Aβ readily forms oligomers with molecular weights (MWs) >50 kDa with spherical, curvilinear, and annular shapes, where the elongated structures appear as "beads-on-a-string"-like assemblies of spherical oligomers[4–11]. While multiple names have been given to these metastable Aβ oligomers, including AβOs, ADDLs, and protofibrils, they seem to be closely related with regard to their structures and detrimental activities and likely form along a common pathway[6,7,12]. Importantly, this pathway is distinct from that of amyloid fibril formation, i.e., AβOs are not intermediates on the pathway to amyloid fibrils (they are "off-pathway") but constitute an alternative Aβ assembly type with distinct toxic activities (Fig. 1a)[4,5,11,13]. The distinct nature of Aβ amyloid fibrils and AβOs is also reflected in their different formation kinetics. Aβ amyloid fibrils form by nucleated polymerization with crucial contributions from secondary nucleation processes, resulting in the characteristic sigmoidal growth time courses that feature an extended lag time[14]. AβOs, on the other hand, form in a lag-free oligomerization reaction that has a substantially higher monomer concentration dependence than amyloid fibril formation[11]. We note that in this work the term AβO refers exclusively to these off-pathway oligomers and does not include other oligomeric Aβ species, such as those transiently formed on the pathway to amyloid fibrils, through secondary nucleation, or through shedding by fibril fragmentation[15].

Several lines of evidence support a critical role of AβOs in AD pathogenesis. AβOs of sizes >50 kDa are the main soluble Aβ species in biological samples[16]. They are synaptotoxic, disrupt long-term potentiation, and cause cognitive impairment in mouse and non-human primate models[4,8,17–23]. Furthermore, AβOs induce oxidative stress, endoplasmic reticulum stress, neuroinflammation, and elicit Tau missorting, the earliest hallmark of tauopathy in AD[21,23–29]. The detrimental effects are enhanced by pathogenic Aβ mutations that specifically promote AβO formation, in particular the arctic (Aβ E22G) and the Osaka (Aβ ΔE22) mutations[22,23,28,30,31]. Consequently, targeting AβOs therapeutically is an important alternative to amyloid-centric approaches and has entered clinical evaluation[32–34].

AβOs were suggested to trigger toxic effects through ligand-like binding to a remarkably high number of candidate receptors[4,35]. AβOs achieve clustering of receptors in cell surface signaling platforms, probably promoted by the multivalency inherent to AβOs[4,35,36]. AβO clustering is especially prominent at dendritic spines, which deteriorate upon prolonged exposure to AβOs[18]. Importantly, this effect is mediated by Tau protein, providing a connection between the Aβ and the Tau aspects of AD pathogenesis. AβOs induce missorting of Tau into the somatodendritic compartment as well as Tau hyperphosphorylation, leading to microtubule destabilization and spine loss[23,37–39].

In addition to receptor binding of extracellular AβOs, intracellular AβOs are thought to contribute to AD pathogenesis[40]. The endosomal–lysosomal system is the main site not only for Aβ production but also for the uptake of Aβ monomers and AβOs[27,41–49]. Aβ accumulates in endosomes/lysosomes, which promotes aggregation with potential consequences for cellular homeostasis as well as for the spreading of Aβ pathology by exocytosis of aggregated Aβ species[27,28,41,44–46,48–51].

At neutral pH, high Aβ concentrations are required to convert a substantial fraction of the protein into AβOs. Widely used protocols for AβO preparation start from around 100 μM Aβ

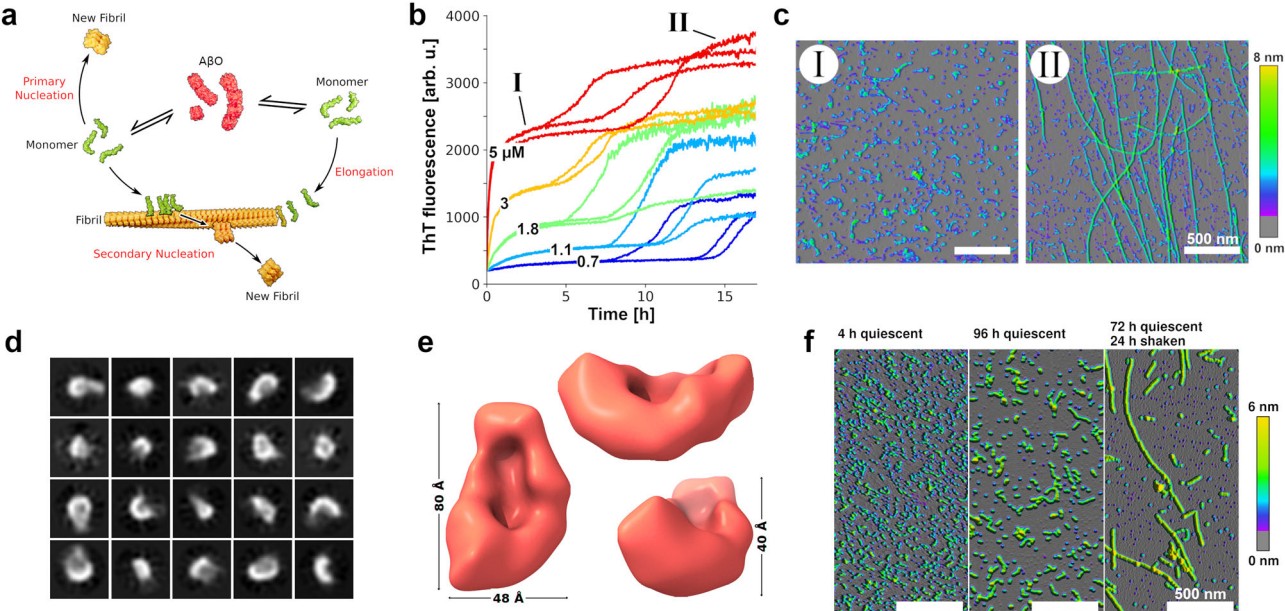

**Fig. 1 AβOs assemble from dimAβ in a lag-free oligomerization reaction. a** Scheme of AβO and amyloid fibril formation. **b** Biphasic assembly kinetics of dimAβ at pH 7.4 and indicated concentrations monitored by ThT fluorescence. The experimental replicates illustrate the good reproducibility of the nucleation-free oligomerization phase and the stochastic nature of the nucleation-dependent fibril growth phase. **c** AFM images corresponding to the two kinetic phases as indicated in **b**. **d** Exemplary 2D classes of the smallest dimAβ AβO species observed in cryo-EM micrographs. **e** 3D density reconstruction of this dimAβ AβO species at a resolution of 17 Å by cryo-EM. The comparatively low resolution is due to the small size and high degree of heterogeneity of the dimAβ AβO species. Consequently, only a rough estimate to size and volume can be made. **f** AFM images of dimAβ assemblies formed upon incubation at pH 7.4 in microcentrifuge tubes. Kinetics data as shown in **b** was obtained from at least three independently prepared assays with two to three replicates for each concentration for reproducibility. AFM images in **c** were prepared from two independent assays and at least three areas at different positions on the mica surface were scanned. The experiment in **f** was done once and at least two sections of the mica surface were scanned.

monomers[7,8,10]. At tenfold lower Aβ concentration, the formation of AβOs is already greatly disfavored, which enables the investigation of the pure sigmoidal time course of amyloid fibril formation, including the analysis of on-pathway oligomer formation[14,15,52]. These on-pathway oligomers, however, are short-lived, rapidly consumed in the process of fibril formation, and, as evident from the different assembly kinetics, clearly distinct from the neurotoxic off-pathway AβOs introduced above. To investigate AβO formation, we have generated a dimeric variant of Aβ termed dimAβ, in which two Aβ40 units are linked in one polypeptide chain through a flexible glycine–serine-rich linker[11]. In dimAβ, the conformational properties of the Aβ40 units are not altered as compared to free Aβ40 monomers[11]. The linkage of two Aβ units, however, increases the local Aβ concentration, which strongly promotes the highly concentration-dependent formation of AβOs[11] (Fig. 1b, c). The advantages in applying dimAβ for the study of AβOs are: First, AβOs form already above a threshold concentration (critical oligomer concentration (COC)) of ~1.5 μM dimAβ at neutral pH. Second, the increased local Aβ concentration preferentially accelerates AβO formation as compared to Aβ fibril formation, resulting in an enhanced separation of the kinetic phases of AβO and Aβ fibril formation, which facilitates analysis.

There is an apparent discrepancy between the obvious pathogenic relevance of AβOs and the high μM Aβ concentrations required for the conversion of a substantial fraction of the protein into AβOs at neutral pH in vitro, which exceeds the estimated picomolar to nanomolar concentrations of extracellular Aβ in normal brain by several orders of magnitude[44]. However, accumulation of Aβ in the endo-lysosomal system was shown to result in micromolar Aβ concentrations in late endosomes and lysosomes[44], suggesting that these acidic vesicles might be the prime sites of AβO formation. Acidic conditions have been reported to accelerate Aβ aggregation[53]. Here we applied dimAβ and Aβ42 to test whether pH reduction from neutral to endolysosomal pH affects AβO formation. We find that endolysosomal pH in fact strongly accelerates AβO formation, whereas amyloid fibril formation is delayed, suggesting that AβO formation is the dominant aggregation process in endosomes/lysosomes. We furthermore show that dimAβ is a disease-relevant model construct for pathogenic AβO formation by demonstrating that dimAβ AβOs target dendritic spines, induce AD-like somatodendritic Tau missorting, and reduce synaptic transmission in terminally matured primary neurons. This indicates that dimAβ-derived oligomers are suitable for the study of downstream mechanistic and neuropathological events in the progression of AD.

## Results

### DimAβ assembles into AβOs that bind to dendritic spines and potently induce Tau missorting.
The assembly kinetics of dimAβ at neutral pH monitored by ThT show a biphasic behavior above a concentration (COC) of ~1.5 μM, with the first phase corresponding to the lag-free oligomerization into AβOs and the second phase reflecting amyloid fibril formation[11] (Fig. 1b, c). DimAβ AβOs are of spherical and curvilinear shape (Fig. 1c) and rich in β-structure[11], in agreement with the characteristics of AβOs formed from Aβ40 and Aβ42 (refs. [4–6,9,13,21]; for atomic force microscopic (AFM) data of AβOs formed from Aβ42, see below). We applied cryogenic electron microscopy (cryo-EM) to further characterize dimAβ AβOs structurally. Structure determination is hampered by the size and shape heterogeneity of AβOs[7,9,10], which is moreover evolving with time, as observed for AβOs formed from Aβ[9] as well as dimAβ[11]. As larger AβOs seem to be assemblies of small spherical structures, our analysis focused

on the small AβOs observed in the micrographs (Fig. 1d, e and Supplementary Figs. 1–3). The fraction of small AβOs was 72 ± 12% in terms of particle number but only ~2–3% in terms of the number of Aβ molecules within AβOs (Supplementary Fig. 1c). The relation between the small and the elongated curvilinear AβOs cannot be inferred from the micrographs. Nevertheless, structure elucidation of the small AβOs could provide insight into a biologically relevant AβO substructure that may furthermore laterally associate and convert into protofibrillar AβOs[54]. We obtained a three-dimensional (3D) density reconstruction (Fig. 1e) at a resolution of 17 Å, which shows a bowl-shaped structure with dimensions of $80 \times 48 \times 40$ Å. From this reconstruction, we were able to calculate the approximate molecular mass that fits into the density to be 62 kDa (Supplementary Fig. 3; see "Methods"). Therefore, the small AβO species, as visible on the micrographs, likely contains six dimAβ monomers (total MW of 60.2 kDa), which corresponds to 12 Aβ40 units. Dodecameric Aβ oligomers were observed before in AβO preparations from synthetic peptide or isolated from AD brain or mouse models and have been associated with neuronal dysfunction and memory impairment[55–58].

AβO formation occurred on the same time scale in the plate reader experiment as in microcentrifuge tubes (Fig. 1b, c, f). In contrast, extensive amyloid formation was observed in the plate reader experiment after ~10 h but was not detectable when AβOs were incubated in microcentrifuge tubes for several days, unless the microcentrifuge tube was agitated (Fig. 1b, c, f). This suggests that the movement of the microplate in the plate reader, caused by scanning of the wells during measurements every 3 min and 2 s of preceding orbital shaking, creates sufficient agitation to promote amyloid fibril nucleation. When the samples in the microplate were covered with a layer of mineral oil, AβO formation was unaffected but amyloid fibril formation was completely abrogated (Supplementary Fig. 4), in line with the essential role of the air–water interface in Aβ amyloid formation in vitro[59]. The strong effects of agitation[14] and air–water interface on Aβ amyloid fibril formation but not on AβO formation confirms again that their assembly mechanisms are different and is in line with the notion that AβO formation does not involve a nucleation step[11,60]. When AβOs, formed by incubation of dimAβ above the COC, were diluted to sub-COC concentrations, they persisted for >24 h, indicating high kinetic stability (Supplementary Fig. 5). We conclude that AβOs formed from dimAβ under quiescent conditions are kinetically stable, not replaced by amyloid fibrils for several days, and can be applied at sub-μM concentrations. DimAβ AβOs may therefore serve as a favorable AβO model.

To test whether dimAβ AβOs cause the same biological effects as reported for AβOs formed from Aβ40 or Aβ42, we investigated their binding to dendritic spines, their direct cytotoxicity, their capacity to induce Tau missorting, and their consequences for neuronal function. AβOs were formed from 20 μM dimAβ and added to primary mouse neurons (days in vitro 15 (DIV15)–22) to a final concentration of 0.5 μM (all dimAβ AβO concentrations given in dimAβ equivalents). One micromolar Aβ40 was used as monomeric control. DimAβ localized to neuronal dendrites both after 3 and 24 h of treatment, where it partially co-localized with dendritic protrusions positive for filamentous actin (stained by phalloidin), which mark synaptic spines (Fig. 2a). In contrast, Aβ40 monomers did not show substantial localization to dendrites (Fig. 2a). Direct cytotoxicity was assessed by analysis of the sizes and shapes of neuronal nuclei upon staining with NucBlue. The fractions of normal and dense nuclei did not change significantly after incubation with dimAβ AβOs (Fig. 2b, c), indicating the absence of direct cytotoxicity, in line with previous reports on AβOs[61].

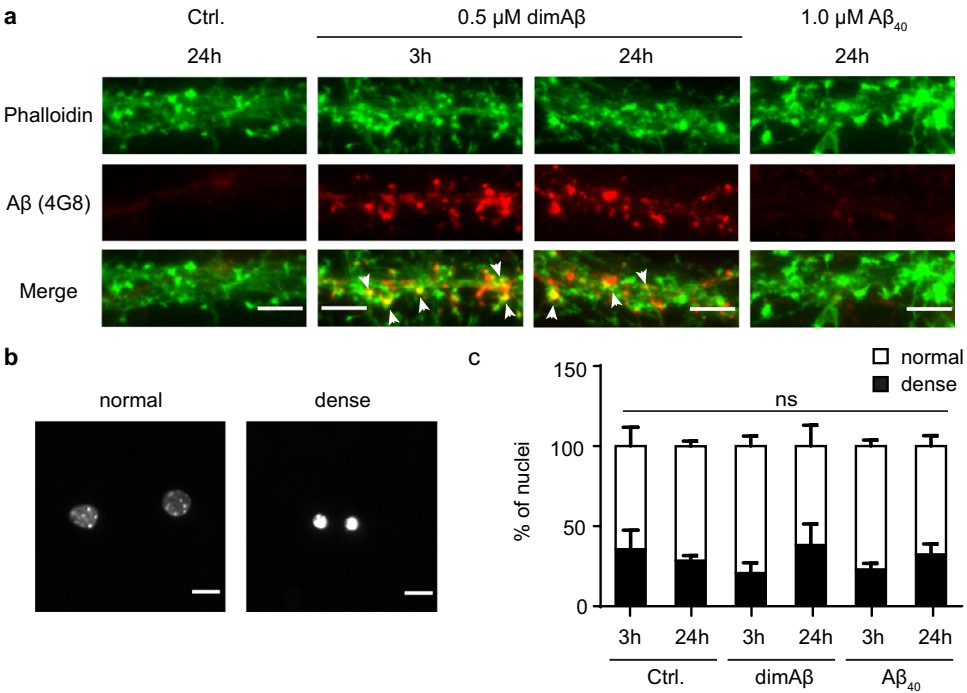

**Fig. 2 DimAβ AβOs bind to dendrites and postsynaptic spines but have no direct cytotoxic effect on primary mouse neurons.** Primary mouse neurons (DIV15–22) were treated with 0.5 μM dimAβ AβOs or 1 μM Aβ40 for 3 and 24 h. **a** DimAβ AβOs localized to neuronal dendrites both after 3 and 24 h of treatment, where they partially co-localized with phalloidin, a marker for synaptic spines. Arrows indicate co-localization of dimAβ with phalloidin. Scale bar, 5 μm. The experiment was independently repeated four times with similar results. **b** Nuclei of primary neurons were stained with NucBlue and analyzed with respect to shape and size. Representative images of normal and dense nuclei. Scale bar, 10 μm. **c** Quantification of normal and dense nuclei of primary neurons after vehicle control, Aβ40, or dimAβ AβO treatment revealed no direct cytotoxicity. $N = 3$; around 300 nuclei were analyzed for each condition. Error bars represent SEM. Statistical analysis was done by two-way ANOVA with Tukey's test for multiple comparisons and yielded no significant differences between the experimental groups.

Tau cellular distribution was analyzed with an anti-Tau (K9JA) antibody. DimAβ AβO-treated neurons showed strong enhancement of the fluorescence signal of Tau in the soma after 24 h of treatment (Fig. 3), indicating pathological somatodendritic Tau missorting as previously reported for AβOs[38,39]. In contrast, Aβ40 monomers did not induce Tau missorting in our experimental setting (Fig. 3). In previous studies, Tau missorting and spine loss were reversible within 12–24 h due to loss of AβO potency (transformation of AβOs over time to larger, non-toxic aggregates)[38,62]. Here we observe an increase of Tau missorting over time, which indicates remarkable kinetic stability and persistent ability of dimAβ AβOs to induce pathological Tau missorting.

Next, we investigated the consequences of AβO exposure for neuronal function. As readout, we measured spontaneous calcium oscillations in our neuronal cultures after dimAβ AβO treatment as an indicator for neuronal activity with live-cell imaging, using the fluorescent cell-permeable calcium indicator Fluo-4 as previously described[38]. A significant decrease of calcium oscillations was observed after 24 h but not after 3 h of treatment with dimAβ AβOs (Fig. 4). As calcium oscillations in our conditions depend on action potentials and neurotransmission, this indicates that dimAβ AβOs impair neuronal activity and function. With regard to dendritic spine binding, lack of direct cytotoxicity, potent induction of Tau missorting as well as decreased neuronal activity, dimAβ AβOs thus faithfully reproduce the observations previously made for AβOs formed from Aβ40 or Aβ42 or from 7:3 Aβ40:Aβ42 mixtures regarded as particularly toxic[38]. Of note, dimAβ AβOs effects appeared later (24 vs. 3 h) than for the previously studied oligomers, hinting toward their kinetic and structural stability in cell culture conditions.

**Aβ42 as well as dimAβ accumulate within endo-lysosomal compartments.** Next, we aimed to test the uptake of dimAβ AβOs in neuronal cells. First, SH-SY5Y neuroblastoma cells were subjected to a mixture of 0.1 μM HiLyte Fluor 647-labeled Aβ42 and 1 μM unlabeled Aβ42. After 24 h of incubation, Aβ42 accumulated within vesicular foci within the cytoplasm of the cells. Co-staining with a LysoTracker dye showed prominent colocalization suggesting the accumulation of Aβ42 within endo-lysosomal compartments (Fig. 5). This is in line with previous studies that showed Aβ42 accumulation in acidic vesicles of neuroblastoma cells and primary murine cortical neurons[41,44–46]. Hu et al. measured local Aβ42 concentrations >2.5 μM within endo-lysosomal compartments, which exceeds the extracellular concentration by approximately four orders of magnitude[44].

In a second attempt, SH-SY5Y cells were treated with 1.1 μM Abberior Star 520SXP-labeled dimAβ AβOs, formed from a mixture of 91% unlabeled and 9% fluorophore-labeled dimAβ (i.e., same final concentrations of unlabeled and fluorophore-labeled Aβ as in the Aβ42 experiment above). This experiment revealed a similar colocalization in acidic vesicles as for Aβ42 (Fig. 5). This confirms that both Aβ monomers and AβOs are readily taken up by neuron-like cells and accumulate in the endo-lysosomal system. Our results, however, do not reveal the assembly state of Aβ, and it is possible that the applied Aβ species undergo structural alterations upon cell entry and accumulation in endo-lysosomes, such as higher-order assembly as described below.

**Endo-lysosomal pH promotes AβO assembly but delays amyloid fibril formation.** Due to the accumulation of Aβ, endosomes/lysosomes might constitute the dominant site of the highly

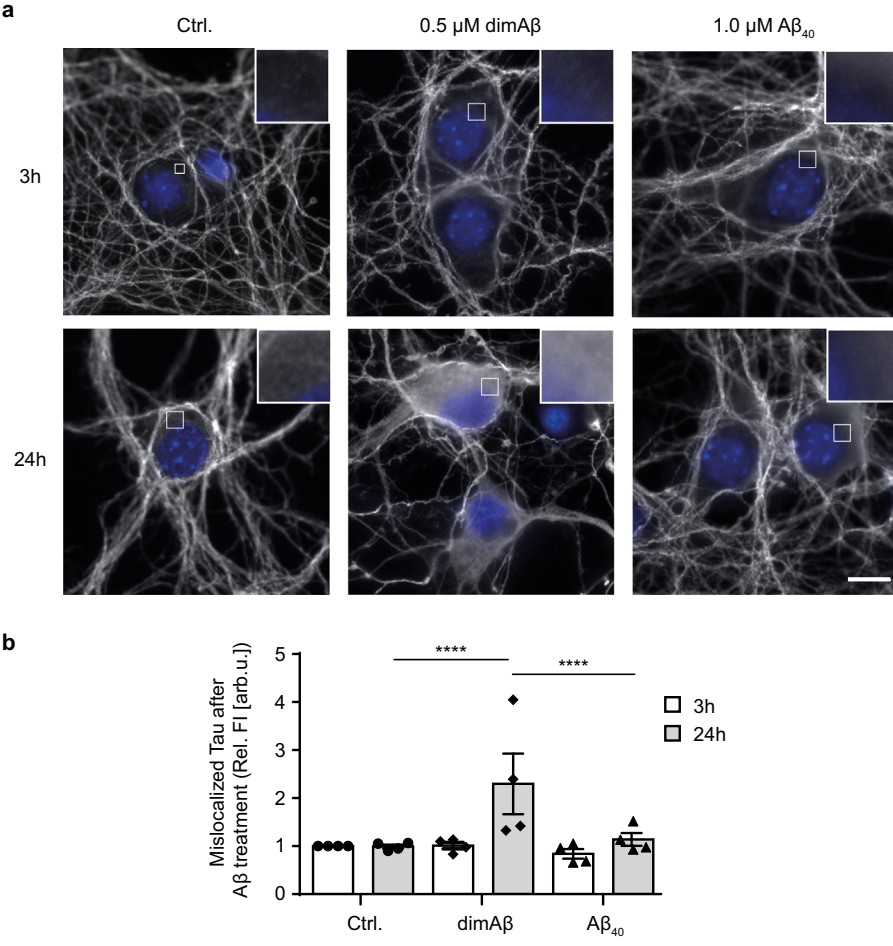

**Fig. 3 DimAβ AβOs induce pathological somatodendritic missorting of Tau.** Primary mouse neurons (DIV15–22) were treated with 0.5 μM dimAβ AβOs or 1 μM Aβ40 for 3 and 24 h. **a** Representative images of cell bodies of primary neurons after treatment with Aβ. Neurons were stained with anti-Tau (K9JA) antibody; nuclei were stained with NucBlue. DimAβ AβO-treated neurons show strong enrichment of fluorescence signal of Tau in the soma only after 24 h of treatment. Insets show magnification of white boxed areas in the somata. Scale bar, 10 μm. **b** Quantification of Tau enrichment in the soma of primary neurons. Fluorescence intensities of cell bodies were quantified and normalized to control-treated neurons after 3 h of treatment. $N = 4$, 30 cells were analyzed for each condition. Error bars represent SEM. Statistical analysis was done by two-way ANOVA with Tukey's test for multiple comparisons. Statistical significance: ****$p < 0.0001$.

concentration-dependent AβO formation. Apart from the increased Aβ concentration in endosomes/lysosomes, the low pH in late endosomes (~5.5) and lysosomes (~4.5) might promote AβO formation. We used dimAβ to simultaneously determine the specific effects of pH on AβO formation and on amyloid fibril formation. Lyophilized dimAβ was dissolved in 6 M buffered guanidinium chloride, followed by size-exclusion chromatography (SEC) into 1 mM NaOH, leading to a pH of 10.9, and added to the wells of a microplate. The basic pH conditions prohibit premature aggregation of Aβ[63]. The pH-dependent aggregation reaction was initiated in the microplate reader by injection of a 10× buffer yielding the desired final pH, allowing for monitoring of ThT fluorescence without any substantial delay. We determined the kinetics of dimAβ assembly between pH 4.8 and 7.6 in the concentration range 0.65–5.0 μM. At neutral pH, the initial kinetic phase reflecting AβO formation spanned several hours, but upon pH reduction, AβO formation was continuously accelerated and occurred within a few seconds at pH 4.8 (Fig. 6a–g). ThT fluorescence intensity decreased at acidic pH[64] but was still sufficiently sensitive to detect the signal of AβO formation at pH 4.8 and 0.65 μM dimAβ (Fig. 6g). For pH 7.4, we have previously shown that a global fit of an $n$th-order oligomerization reaction to the concentration-dependent assembly kinetics is in good agreement with the data and yields a reaction order of ~3.3 for

dimAβ AβO formation[11]. Here we found that a reaction order of three applied to global fitting of the concentration-dependent data results in fits that reproduce the kinetic traces at all pH values (Fig. 6a–g). This indicates that the fundamental mechanism of AβO formation is not affected by pH reduction. A logarithmic plot of the obtained oligomerization rate constants against pH shows a linear trend with a slope of −1.56, i.e., the rate constant decreases 36-fold per pH unit within the investigated pH range (Fig. 6h). At pH 4.8, in between lysosomal and endosomal pH, AβO formation is 7900-fold faster than at interstitial pH (7.3).

In order to test whether the acceleration of AβO formation kinetics is accompanied by thermodynamic stabilization, we evaluated the effect of pH reduction on the COC of dimAβ. In the AβO formation assay at pH 7.4, the fluorescence intensity increase during the lag-free oligomerization phase scaled linearly with protein concentration at dimAβ concentrations above ~2 μM, whereas no lag-free oligomerization was detectable below ~0.5 μM, indicative of a COC of around 1 μM (Supplementary Fig. 6a, b). At pH 5.6, however, there is no indication of disappearance of the oligomerization phase down to a concentration of 0.4 μM dimAβ (Supplementary Fig. 6c, d). Due to the limited sensitivity of ThT at acidic pH[64], it is not possible to reliably monitor oligomerization at lower concentrations and to

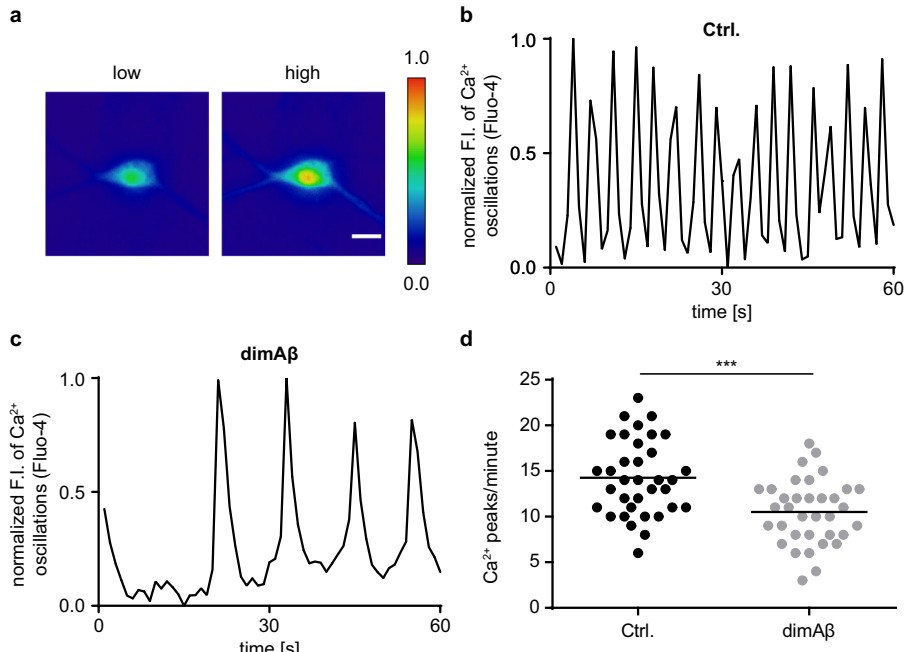

**a**

low high

1.0

0.0

**b** Ctrl.

**c** dimAβ

**d**

**Fig. 4 DimAβ AβOs decrease spontaneous calcium oscillations of primary mouse neurons.** Primary mouse neurons (DIV15–22) were treated with 0.5 µM dimAβ AβOs for 24 h. Cells were labeled with calcium-sensitive Fluo-4 dye and spontaneous calcium oscillations were recorded by time-lapse movies. **a** Representative ratiometric images of low and high calcium concentrations in the soma of a neuron. Scale bar, 20 µm. **b**, **c** Representative graphs of spontaneous $Ca^{2+}$ oscillations in **b** vehicle control- and **c** dimAβ AβO-treated primary neurons. Fluorescence intensities were normalized to minimum values and plotted over time. **d** Quantification of spontaneous $Ca^{2+}$ oscillations in primary neurons after vehicle control or dimAβ AβO treatment. Fluorescence intensities were normalized to minimum values and peaks per minute were counted for each sample. In total, 35 cells were analyzed; statistical analysis was done by two-tailed unpaired t test. Statistical significance: ***$p = 0.0001$.

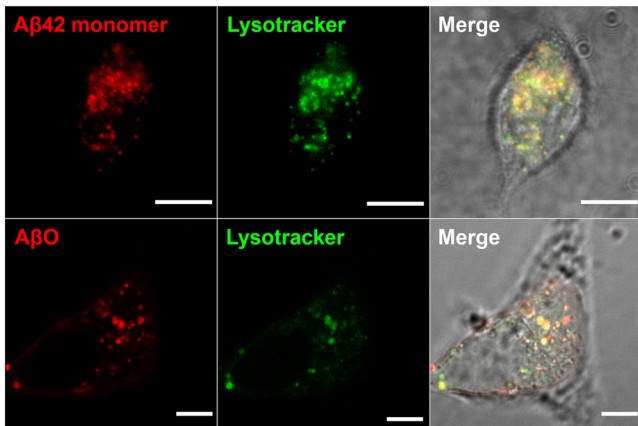

Aβ42 monomer Lysotracker Merge

AβO Lysotracker Merge

**Fig. 5 Aβ42 and dimAβ AβOs accumulate in endosomes/lysosomes.** SH-SY5Y cells were treated with Aβ42 monomers (top row) or dimAβ AβOs (bottom row) and co-localization with endo-lysosomal compartments was analyzed. 1.1 µM Aβ42 (containing 9% HiLyte 647-labeled Aβ42, top row) or 1.1 µM dimAβ AβOs (in monomer equivalents, formed from a dimAβ solution containing 9% AbberiorStar 520SXP-labeled dimAβ, bottom row) were added to the cells. After 24 h, the medium was exchanged with fresh medium supplemented with 50 nM Yellow HCK-123 LysoTracker dye. Scale bar, 5 µm. N = 3, at least three images were acquired for each treatment to ensure reproducibility.

determine the COC at this pH. Nevertheless, the COC at pH 5.6 is clearly lower than the COC at neutral pH, indicative of thermodynamic stabilization of AβOs at acidic pH.

AβOs formed at different pH values were imaged by AFM (Fig. 6i–o). From pH 7.6 to pH 6.8, AβOs were mainly spherical and curvilinear structures, the latter apparently resulting from

bead-chain-like association of the spherical AβOs[6]. At pH 6.4, AβOs showed an increased tendency to form more compact structures, such as annular protofibrils and denser clusters. Below pH 6.0, AβOs associated into large clusters, in line with a previous description of Aβ40 aggregates at pH 5.8[53]. In AFM, these AβO clusters have average heights of ~100 nm, compared to heights of ~4 nm observed for AβOs formed between pH 6.0 and 7.2 (Fig. 6p). Thus, while the fundamental mechanism of AβO formation seems to be unaffected by pH reduction, there is an additional level of particle aggregation involved below pH 6.0.

The second kinetic phase in the ThT time course of dimAβ aggregation reports on amyloid fibril formation[11]. It is characterized by a lag time, which reflects the primary and secondary nucleation events involved in nucleated polymerization[14,52]. In contrast to the acceleration of AβO formation, the lag time of amyloid formation did not decrease with decreasing pH. On the contrary, the amyloid fibril formation phase could not be observed within 10 h experiments at pH values of 6.8 and below. This can be explained by the inhibition that the rapidly forming AβOs entail on amyloid formation: First, AβOs compete for the monomer growth substrate of amyloid fibril growth; second, AβOs actively inhibit amyloid fibril growth[11,65].

**AβO assembly of Aβ42 is enabled under endo-lysosomal conditions.** We investigated whether the promotion of AβO formation at endo-lysosomal pH is sufficient to also support AβO formation from Aβ42 at relevant endo-lysosomal Aβ concentrations, determined to be well above 2.5 µM[44]. At pH 7.2, Aβ42 in the concentration range 1.9–9 µM displayed sigmoidal assembly kinetics typical for amyloid fibril formation (Fig. 7a). The absence of a lag-free oligomerization phase is in agreement with the observation that the COC of Aβ42 in in vitro assay at neutral pH

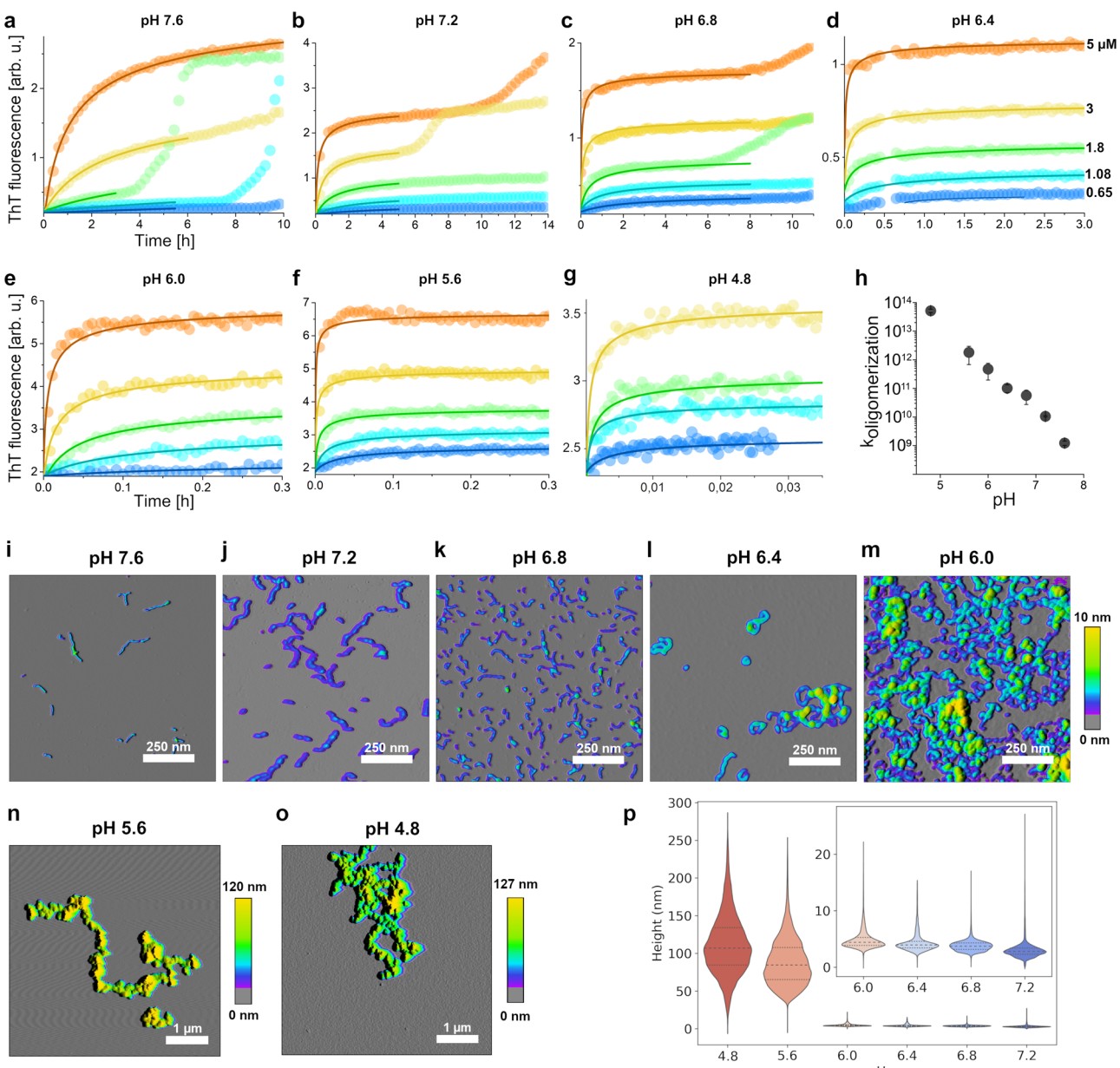

**Fig. 6 pH dependence of dimAβ assembly kinetics. a–g** DimAβ assembly at concentrations between 0.65 and 5 μM and at pH values between 4.8 and 7.6 monitored by ThT fluorescence. Solid lines represent global fits to the data using a one-step oligomerization model with a shared reaction order of 3 for all pH values and concentrations and an individual oligomerization rate constant per pH value. **h** Logarithmic plot of the obtained oligomerization rate constants vs. pH. The rate constants were obtained from global fits to *n* concentration dependence data sets obtained from *m* independently prepared assays, with *n/m* being 2/2 (pH 4.8), 6/4 (pH 5.6), 8/4 (pH 6.0), 5/4 (pH 6.4), 6/2 (pH 6.8), 6/2 (pH 7.2), and 6/2 (pH 7.6). One of the *n* repeats is shown in **a**–**g**. Replicates are given in Supplementary Fig. 8. Data points represent mean and standard deviation, except for pH 4.8, where the error bar indicates the higher and lower value of the *n* = 2 experiments. **i–o** AFM images of dimAβ AβO formed at different pH values. Note the dramatic change in the height scale bar upon pH decrease to <6.0 due to formation of large AβO clusters. Between 7 and 25 micrographs of at least 2 independent assays were recorded for each pH value to ensure reproducibility. **p** Particle height distributions determined from AFM images, displayed as violin plots. All pixels assigned to AβOs by the image analysis software in five micrographs per pH value were evaluated. Dashed lines represent medians; dotted lines represent interquartile ranges. Inset, zoom on the data for pH 6.0 to pH 7.2.

is >10 μM[65]. Consequently, the aggregation products under this condition are amyloid fibrils (Fig. 7c, f). In contrast, at pH 4.5 lag-free aggregation occurred at a concentration of ≥5.4 μM (Fig. 7b). The change from lag-containing to lag-free conditions at pH 4.5 was accompanied by a switch in aggregate morphology from amyloid fibril networks to large AβO clusters identical to those observed for dimAβ at endo-lysosomal pH (Fig. 7d, e, g, h). This indicates that under endo-lysosomal conditions the local Aβ concentration can exceed the COC of AβO formation, suggesting

that endosomes/lysosomes may represent crucial sites of AβO formation in vivo.

Aβ aggregates can leak from endosomes/lysosomes into the cytosol and to other cell compartments or can be secreted and spread to other cells, potentially contributing to the propagation of Aβ pathology[27,28,44,45,51]. Upon transfer from endosomes/lysosomes to the cytosol or interstitial fluid, AβOs experience a shift from acidic to neutral pH. We tested the kinetic stability of AβOs formed at pH 4.5 after a shift to neutral pH by monitoring

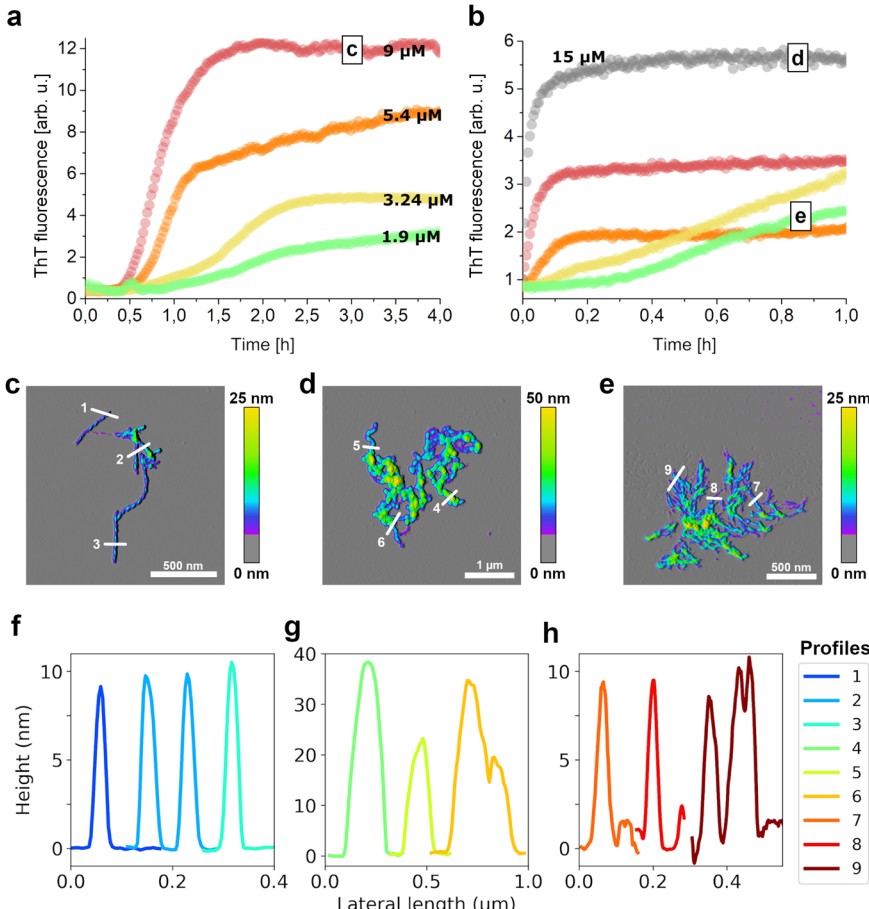

**Fig. 7 Aβ42 rapidly forms AβOs at endo-lysosomal pH. a**, **b** Aβ42 assembly at **a** pH 7.2 or **b** pH 4.5 at concentrations between 1.9 and 15 μM monitored by ThT fluorescence. Replicates are given in Supplementary Fig. 9. **c–e** AFM images of **c** amyloid fibrils formed by 9 μM Aβ42 at pH 7.2, **d** AβOs formed by 15 μM Aβ42 at pH 4.5, and **e** amyloid fibril networks formed by 1.9 μM Aβ42 at pH 4.5. At least three micrographs each of two independently prepared sample repeats were recorded to ensure reproducibility of the AFM data. **f–h** Height profiles of the sections indicated in **c–e**.

the ThT intensity and by imaging of the aggregate morphology by AFM. We applied Aβ42 at a concentration of 10 μM in this experiment, as Aβ42 does not form AβOs de novo at this concentration at neutral pH. Any AβOs observed after the pH shift can therefore safely be ascribed to the kinetic stability of AβOs pre-formed under acidic conditions. As before, a pH shift from basic pH to pH 4.5 was applied to initiate AβO formation. After AβO formation had reached a steady state, pH was adjusted to 7.2 by a further injection of a corresponding buffer stock. After the adjustment to neutral pH, there was an instantaneous increase in ThT fluorescence (Supplementary Fig. 7), which can be explained by the pH dependence of ThT fluorescence[64]. Thereafter, the ThT fluorescence did not exhibit any other larger changes that would be expected in the case of disassembly of AβOs or replacement of AβOs by an alternative type of aggregate. Apart from dense clusters like those observed for low pH AβOs, AFM images showed spherical and curvilinear structures typical for AβOs formed at neutral pH, indicating dissociation of the AβO clusters into their constituents (Fig. 8a). In fact, the AFM images suggest that smaller AβOs detach from fraying AβO clusters. The height of the cluster-released Aβ42 AβOs was 3.5–4.5 nm as measured by AFM in the dried state (Fig. 8b, c), identical to that of Aβ42 AβOs (Fig. 8d, e) and dimAβ AβOs (Fig. 6p) that were directly formed at neutral pH. Taken together, the ThT and AFM data demonstrate that AβOs formed at endo-lysosomal pH possess a high kinetic stability after shifting to

neutral pH, which is, however, accompanied by dissociation of large AβO clusters into spherical and curvilinear AβOs.

## Discussion

AβOs have been identified as the main neurotoxic Aβ species in AD. The characterization of the most critical disease-related AβOs has revealed that they are metastable oligomers >50 kDa in size that do not represent intermediates of amyloid fibril formation but are an alternative Aβ assembly type. However, the conditions required for AβO formation and the underlying mechanism have not been elucidated in detail. Here we show that AβO formation is highly pH dependent and is accelerated ~8000-fold upon a change in pH from neutral to endo-lysosomal pH. At the same time, the COC of AβO formation is reduced. This enables AβO formation at physiologically relevant Aβ concentrations, determined to be well above 2.5 μM in endo-lysosomal vesicles[44]. The strong acceleration of AβO formation at pH 4.5–5.5 suggests that the endosomal/lysosomal system might be a major site of AβO formation. AβOs may either form from Aβ monomers that have been newly generated by amyloid precursor protein (APP) processing or from endocytosed monomers (Fig. 9)[40–42,44,47,48]. APP processing in endo-lysosomal compartments by γ-secretase containing presenilin 2 generates a prominent pool of intracellular Aβ that is enriched in Aβ42 (ref.[48]). Esbjörner et al. applied fluorescence lifetime and

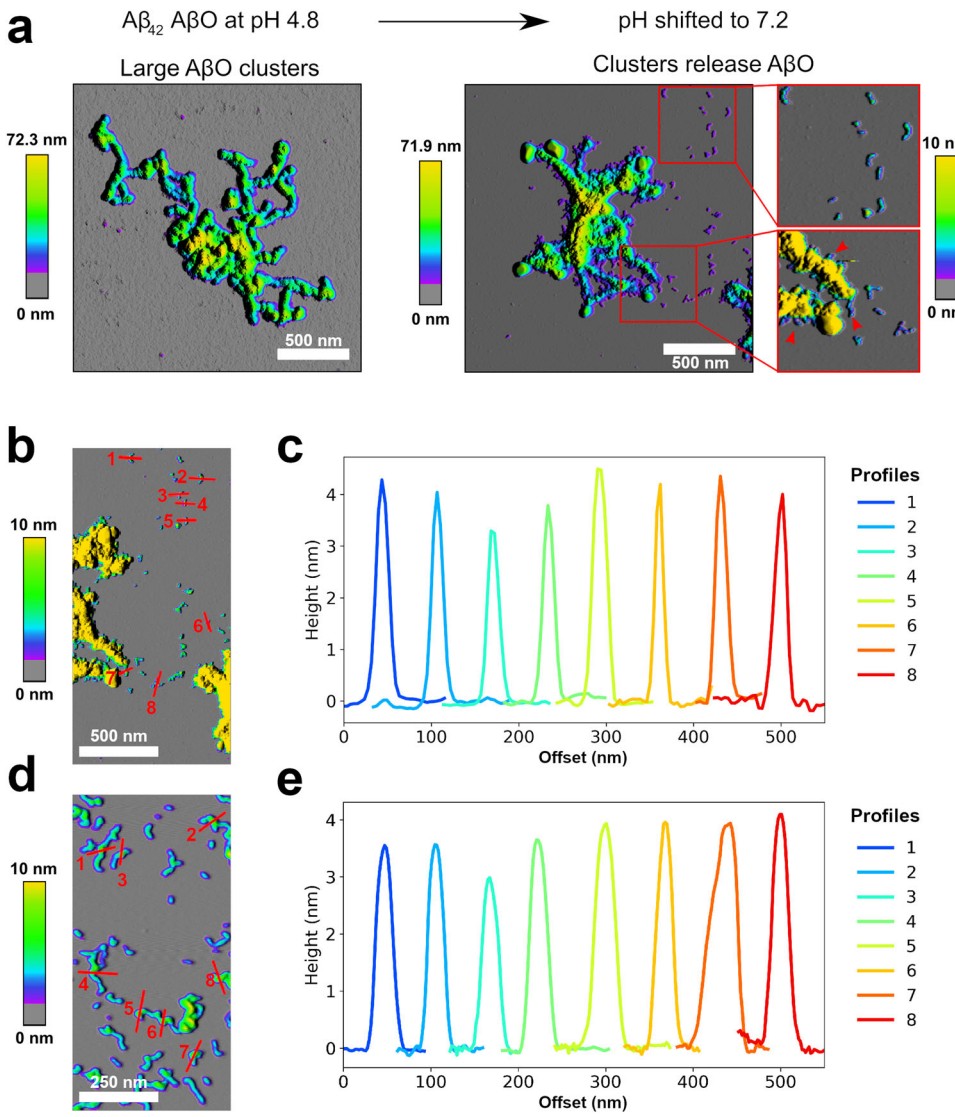

**Fig. 8 Stability of AβOs formed by Aβ42 at endo-lysosomal pH after shifting to neutral pH. a** AFM images of AβOs formed by 10 μM Aβ42 at pH 4.5 before (left) and after (right) shift to pH 7.2. Red arrowheads point to a few of the sites where AβOs seem to detach from AβO clusters. In all, 3–7 micrographs were recorded per condition to ensure reproducibility. **b**, **c** Height profiles of small AβOs after pH shift to neutral pH. Height profiles in **c** correspond to the sections in **b**. **d**, **e** Height profiles of AβOs formed by 110 μM Aβ42 at pH 7.2. Height profiles in **e** correspond to the sections in **d**.

super-resolution imaging to determine the kinetics of Aβ aggregation in live cells and found that aggregation occurred in endo-lysosomal compartments[41]. Importantly, they reported that Aβ42 aggregated without a lag time into compact, dense structures[41]. Both the absence of a lag time and the structural characterization are in line with the low pH AβO clusters described here, suggesting that AβO clusters indeed form in endo-lysosomal compartments and represent the dominant Aβ aggregate species in live cells. Subsequently, AβOs might cause lysosomal impairment, leak into the cytosol and cause intracellular damage, or might be secreted and spread to neighboring cells, where they could contribute to the propagation of pathology[40,42,44–46,48,49].

Enhanced aggregation at acidic pH is a known property of Aβ with established relevance for sample preparation[63]. Our results are in line with a study on the aggregation of Aβ40 (at a concentration of 230 μM) at pH 5.8 that reported the rapid formation of large clusters with (proto)fibrillar and globular substructures that were not able to seed, but rather inhibited, amyloid fibril formation[53]. Our analysis of the aggregation kinetics reveals that these low pH Aβ aggregates, often termed amorphous aggregates,

form along the same pathway as neutral pH AβOs and therefore represent particle aggregates of AβOs. This is supported by the observation that low pH AβO clusters release spherical and curvilinear AβOs upon a shift to neutral pH (Fig. 8a). Nevertheless, there may be differences between atomic-level structures and between intermolecular interactions in AβOs formed at different pH, just as atomic-level structures and protofilament interfaces of amyloid fibril polymorphs can differ significantly.

The increasing clustering of AβOs upon pH reduction from neutral to pH 6 points to the high propensity of AβOs to associate. At neutral pH, self-association of spherical AβOs results in curvilinear assemblies. A decrease of pH leads to an increase in annular and compact assemblies and finally to large AβO clusters (Fig. 6). This propensity of AβOs to associate likely also contributes to their clustering with neuronal receptors[35,36] and to their accumulation around amyloid fibril plaques[66].

In contrast to AβO formation, amyloid fibril formation of dimAβ is slowed down at acidic pH. This pH dependence is not an inherent property of Aβ amyloid fibril formation: in the absence of AβOs, Aβ42 amyloid fibril formation occurs rapidly at

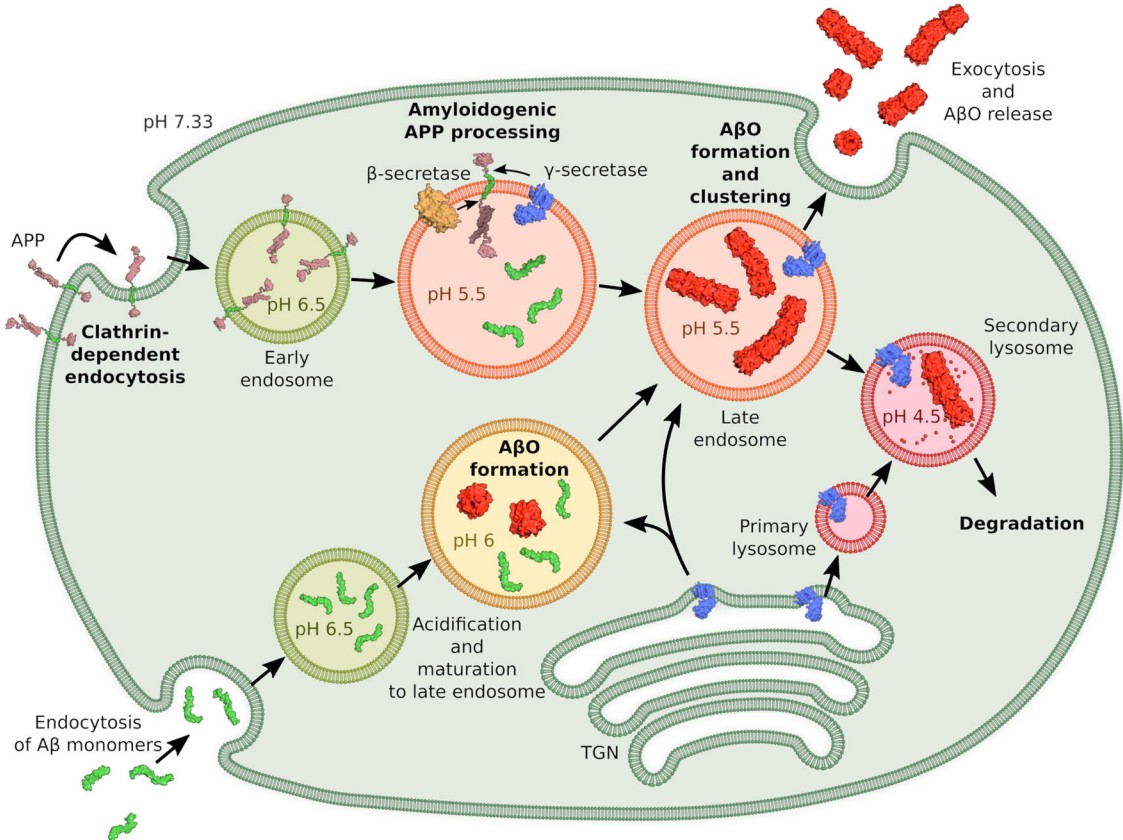

**Fig. 9 Scheme of intracellular APP processing, Aβ uptake, and AβO formation.** This is an extension of previous schemes of APP processing and Aβ uptake[48,76,77], now including potential formation of AβOs especially in endo-lysosomal compartments. Using a conservative estimate of the endo-lysosomal Aβ concentration of 2.5 μM[44] and assuming an endosome volume of 0.3 μm$^3$, there are on average 450 Aβ molecules in an endosome. Protein structure images were prepared using pdb entries 1OWT, 1IYT, 1RW6, 3DXC, 4UIS, and 1SGZ. TGN trans-Golgi network.

pH 4.5 (Fig. 7b, 1.9 μM trace). Delayed amyloid fibril formation upon pH reduction is only observed in combination with accelerated AβO formation and can be explained by the two inhibitory activities of AβOs on amyloid fibril formation: AβOs compete with amyloid fibrils for monomers (Fig. 1a) and furthermore inhibit amyloid fibril growth actively[11,65].

DimAβ AβOs show dendritic spine binding, lack direct cytotoxicity, potently induce Tau missorting, and decrease neuronal activity, suggesting that they constitute a suitable AβO model construct to study the pathomechanism of AD. Previous AβO preparations showed a loss of potency to induce Tau missorting within 12 h due to transformation to non-toxic larger Aβ aggregates[38,62]. In contrast, dimAβ AβOs led to extensive and persistent Tau missorting 24 h after application. The sustained activity of dimAβ AβOs is likely a consequence of the kinetic stabilization of the AβO state achieved by the dimer linkage. DimAβ might therefore be an advantageous model for eliciting Tau missorting and downstream consequences, as it represents a model of chronic stress corresponding to the human disease rather than acute insult.

## Methods

**Preparation of dimAβ.** DimAβ was produced recombinantly[11]. Expression of dimAβ was achieved by co-expression of ZAβ3, a binding protein that shields aggregation-prone sequence segments of Aβ[67]. The gene encoding dimAβ included an N-terminal methionine, followed by a Aβ40 unit, a (G$_4$S)$_4$ linker, and a second Aβ40 unit. DimAβ and (His)$_6$-tagged ZAβ3 were co-expressed from a pACYCDuet-1 vector that contained the genes in the following order: T7promoter-1–dimAβ–T7promoter-2–(His)$_6$ZAβ3–T7 terminator. BL21(DE3) E. coli cells (Novagen) were transformed with the expression vector and grown for ~16 h at 37 °C on LB agar plates containing 34 μg/ml chloramphenicol. Single colonies were

picked and grown for ~16 h in 50 ml M9 medium, containing 2×YT medium and 34 μg/ml chloramphenicol. In all, 40 ml of the pre-culture was transferred to 2 l of M9-Celtone medium in a 5 l baffled Erlenmeyer flask. The culture was grown at 37 °C with shaking and induced at OD$_{600}$ ~ 0.8 by the addition of IPTG to a final concentration of 1 mM. After further growth for 4 h, the cells were harvested and frozen at −20 °C.

For purification, cell pellets were resuspended in 50 mM Na-phosphate, 0.3 M NaCl, 20 mM imidazole, pH 8, containing EDTA-free protease inhibitor (Roche Applied Sciences), and lysed by a cell disrupter (Constant Systems). The cell debris was removed by centrifugation in a Beckman J2-21 centrifuge mounting a JA20.1 rotor at 18,000 RPM, 4 °C for 40 min. For capture of the dimAβ:ZAβ3 complex by immobilized metal ion affinity chromatography (IMAC), the supernatant was loaded on a HisTrap FF column (GE Healthcare). DimAβ was separated from the resin-bound ZAβ3 and eluted with 8 M urea and 20 mM Na-phosphate, pH 7. For further purification, including removal of residual ZAβ3, reverse-phase high-performance liquid chromatography (RP-HPLC) was performed. For this purpose, the IMAC eluate was concentrated in a Vivaspin 20 centrifugal concentrator (Sartorius), followed by addition of 5 mM TCEP to reduce the disulfide bond of ZAβ3, and loading onto a semi-preparative Zorbax 300SB-C8 RP-HPLC column (9.4 mm × 250 mm, Agilent) connected to an Agilent 1260 Infinity system with UV detection at 214 nm. Monomeric dimAβ was eluted in a gradient from 30% (v/v) to 36% (v/v) acetonitrile in water and 0.1% (v/v) trifluoroacetic acid at 80 °C. DimAβ-containing fractions were pooled, lyophilized, dissolved in hexafluoroisopropanol (HFIP), aliquoted in 1 mg portions, lyophilized again, and stored at −20 °C.

For aggregation kinetic experiments, the lyophilized protein was reconstituted in 6 M guanidinium chloride and 50 mM sodium-phosphate buffer, pH 7.4, and incubated at room temperature (RT) for 30 min. Subsequently, SEC was performed using a Superdex 75 increase column (GE Healthcare) equilibrated with 1 mM NaOH. The concentration of the monomeric dimAβ in the alkaline eluate was measured via tyrosine fluorescence using a pH-adjusted extinction coefficient of 2685 M$^{-1}$ cm$^{-1}$. Samples were always kept on ice until further needed.

**ThT aggregation kinetics.** ThT, NaN$_3$, NaCl, protein, and 1 mM NaOH were given into the wells of a 96-well low-binding plate (Greiner) such that if filled up to 100 μl, concentrations of 1 μM ThT, 0.02% NaN$_3$, 150 mM NaCl, and the desired final protein concentration were reached. The outermost wells of the plate were left

blank due to the risk of aberrant aggregation behavior. The plate was put in a BMG ClarioStar platereader fitted with two injectors and tempered at 37 °C. One syringe of the injector was equilibrated with 1 ml 10× buffer concentrate. The reaction was started using the injector of the platereader by dispensing 10 μl of the concentrate at highest available speed into each of the wells. This adjusted the pH value in situ and initiated oligomerization. Data points were collected in evenly spaced intervals depending on the velocity of the reaction using the BMG Reader Control software (version 5.40).

For shifting the pH in situ twice, both syringes were equilibrated with 10× buffer concentrate; the first one resulting in a final buffer concentration of 20 mM and pH 4.5 and the second one resulting in a final buffer concentration of 50 mM and pH 7.2. The first syringe was used to inject 10 μl to initiate oligomerization, whereas the second one was used to inject 11 μl to achieve the shift to neutral pH at a time point where the oligomerization reaction had reached its plateau.

For analysis of the kinetics of AβO formation, the initial phase of the ThT kinetics was fit to one-step oligomerization $nM \rightarrow M_n$ (ref. [11]). The AβO mass concentration, $M_{A\beta O}$, evolves in time according to the following expression

$$M_{A\beta O}(t) = M_0 - [M_0^{1-n} + (n-1)nkt]^{1/(1-n)} \quad (1)$$

with $M_0$ the total protein concentration, $k$ the oligomerization rate constant, and $n$ the oligomer size or reaction order. Global fits to the pH- and concentration-dependent AβO formation data were performed using the Origin 9.0 software with a reaction order of $n = 3$ shared between all data sets, and the oligomerization rate constant $k$ as a pH-dependent parameter, which was shared within the concentration-dependency data sets at a given pH. The proportionality constant relating $M(t)$ to ThT fluorescence intensity was treated as a fit parameter with an individual value for every sample.

**Atomic force microscopy.** In all, 10 μl of the dimAβ samples were taken directly from the plate after the ThT assays at a concentration of 5 μM and applied onto freshly cleaved muscovite mica. They were left to dry, washed with 500 μl ddH₂O, and dried with a stream of N₂ gas. For imaging dimAβ at pH 4.8, the aforementioned method did not work, likely due to sticking of the sample to the well. Instead, all reaction components apart from the buffer concentrate were premixed and loaded into a micropipette tip. By adding the reaction components to a vial containing the buffer concentrate and thorough mixing, the reaction was started, before pulling the solution back into the tip. Immediately afterwards, the micropipette was relocated into a 37 °C incubation cabinet, where a drop was pushed out to the point where it still stuck to the tip. After 45 s, the drop was pushed onto the freshly cleaved muscovite mica and preparation commenced as with the other pH values.

For the Aβ42 samples, 5 μl of the respective concentrations were taken, applied onto freshly cleaved muscovite mica, and left to dry for 15 min before carefully washing with 200 μl ddH₂O and drying under a stream of N₂ gas.

Imaging was performed in intermittent contact mode (AC mode) in a JPK Nano Wizard 3 atomic force microscope (JPK, Berlin) using a silicon cantilever with silicon tip (OMCL-AC160TS-R3, Olympus) with a typical tip radius of 9 ± 2 nm, a force constant of 26 N/m, and resonance frequency around 250 kHz. The images were processed using the JPK DP Data Processing Software (version spm-5.0.84). For the presented height profiles, a polynomial fit was subtracted from each scan line first independently and then using limited data range. False-color height images were overlaid onto the amplitude profile.

Particle height distributions were extracted from AFM images. Therefore, the Morphological Active Contours without Edges (MorphACWE) function of python's scikit-image module was used to distinguish and separate AβOs from background (see Supplementary Fig. 10 for examples of AFM image segmentation). Histographical height profiles of AβOs at different pH were determined as per pixel heights of the MorphACWE-isolated areas.

**Cryo-EM.** For cryo-EM imaging, the AβO sample was plunge-frozen on glow-discharged Quantifoil 1.2/1.3 grids. In total, 1308 micrographs were recorded as focal pairs at high defocus (6 μm) and low defocus (using a range of −0.5 to −2 μm) on a Tecnai Arctica (200 kV) using a Falcon III direct electron detector, yielding a pixel size of 0.935 Å. Particle selection was performed automatically using crYOLO[68]. In total, 32,211 particles were selected on the high defocus micrographs. The contrast transfer function of the micrographs was determined using CTFFIND4[69]. Further image processing was performed using the software package RELION 3.0.5[70]. Two-dimensional and 3D classification was conducted on the high-defocus images to clean the data set. A box size of 128 pix, which corresponds to 119.7 Å, and a radial mask with a diameter of 100 Å were used.

The high-defocus micrographs were aligned to the low-defocus micrographs. The relative shifts obtained from this alignment were applied to all particles (that were picked from the high-defocus micrographs) and then the particles were extracted from the low-defocus micrographs with the shifted particle coordinates, while keeping the Euler angles from the high-defocus 3D refinements. A 3D reconstruction calculated from the high-defocus images was low-pass filtered to 60 Å and was used as an initial model for further low-defocus 3D refinements. For further processing steps, only micrographs that contain a signal beyond a resolution of 5 Å were used. The final resolution of 17 Å was assessed by Fourier shell correlation.

In order to obtain an estimate for the molecular mass within the reconstructed density, 110 pseudo-atomic models with varying number of pseudo-atoms (molecular masses between 10 and 120 kDa) were generated from the density map using the program VISDEM[71], which is part of the software package DireX[72]. In VISDEM, atoms are randomly placed into a density region with density above a provided threshold. The density threshold was set to yield a volume such that the mass density is fixed at 0.714 ml/g (average mass density observed in proteins). The pseudo-atomic model has a composition of 62.2% C atoms, 20.6% O atoms, and 17.2% N atoms, which corresponds to the average composition observed in proteins. Afterwards, a density map was computed from each of the 110 pseudo-atomic models. The VISDEM method was used to sharpen these pseudo-atomic model maps as well as the EM reconstruction. The sharpening was performed with a resolution cutoff of 17 Å and the mass of the corresponding pseudo-atomic model. Finally, the cross-correlation between the sharpened EM reconstruction and the sharpened pseudo-atomic model map was computed and plotted for each tested model. The highest cross-correlation was found for the pseudo-atomic model map that contains a molecular mass of 62 kDa. One dimAβ monomer (101 amino acids) has a molecular mass of 10.0 kDa. Thus, the reconstructed density likely holds six dimAβ monomers. The final 3D reconstruction of the oligomer was sharpened by VISDEM using a mass of 62 kDa and a resolution cutoff of 17 Å.

**Preparation of dimAβ AβOs and Aβ40 monomers for treatment of primary neurons.** Aβ preparations were performed under sterile conditions. DimAβ lyophilisate was resuspended in 50 mM NaOH until completely dissolved. Next, phosphate-buffered saline (PBS) and 50 mM HCl were added and immediately mixed, obtaining a final concentration of 20 μM dimAβ and 40 μM Aβ40. To induce AβO formation, dimAβ was incubated at 37 °C for 16 h. Aβ40 controls were prepared in the same manner without subsequent incubation. Primary neurons (DIV15–22) were treated with either 0.5 μM dimAβ AβO or 1 μM Aβ40 monomers diluted in conditioned neuronal maintenance media for 3 and 24 h under normal growth conditions (see below). In addition, control cells were treated with a vehicle control (PBS containing 50 mM NaOH and 50 mM HCl). Afterwards, cells were fixed and stained as described below.

**Primary neuron culture.** Primary neurons were isolated and cultured as described before[73] with slight modifications: In brief, the brains of FVB/N mouse embryos were dissected at embryonic day 13.5. Brainstem and meninges were removed and whole cortex was digested with 1× Trypsin (Panbiotech). Neurons were diluted in pre-warmed (37 °C) neuronal plating medium (Neurobasal media (Thermofisher Scientific), 1% fetal bovine serum (FBS; Biochrom AG), 1× antibiotic/antimycotic solution (Thermofisher Scientific), 1× NS21 (Panbiotech)) and seeded onto poly-D-lysine (Merck) coated coverslips. Neurons were cultivated in a humidified incubator at 37 °C, 5% CO₂. Four days after plating, media was doubled with neuronal maintenance media (Neurobasal media (Thermofisher Scientific), 1× antibiotic/antimycotic solution (Thermofisher Scientific), 1× NS21 (Panbiotech)) and cells were treated with 0.5 μg/ml Cytosine β-D-arabinofuranoside (AraC; Sigma-Aldrich). The isolation of primary neurons was reviewed and approved (§4 TschG) by the Animal Welfare Officer of University of Cologne and the Landesamt für Natur-, Umwelt- und Verbraucherschutz (LANUV), Germany.

**Somatodendritic missorting of Tau.** To analyze Tau somatodendritic localization, neurons were fixed with 3.7% formaldehyde/4% sucrose in PBS (both Sigma-Aldrich) for 30 min at RT using gentle agitation after treatment with Aβ or vehicle control for the indicated time points. Afterwards, cells were permeabilized and blocked for 5–10 min in 5% bovine serum albumin/0.2% TX-100 in PBS (both Carl Roth), washed with PBS, and stained with a polyclonal rabbit anti-Tau (K9JA, Dako A0024; dilution: 1:1000) antibody overnight at 4 °C. The next day, coverslips were washed again with PBS, incubated with NucBlue (Thermofisher Scientific) for 15 min, and subsequently stained with a secondary antibody coupled to an AlexaFluor dye (Thermofisher Scientific) for 1 h at RT. Coverslips were mounted onto glass slides using Aqua-Poly/Mount (Polysciences) and dried overnight at RT (for further details on immunofluorescence staining procedure, see ref. [73]). Images of neuronal cell bodies were taken with a wide-field fluorescence microscope (Axioscope 5, Zeiss) and the ZenBlue Pro imaging software (V2.5, Zeiss). Fluorescence intensities of cell bodies were quantified using the ImageJ software[74,75]. Fluorescence intensity values were normalized to vehicle-treated control cells after 3 h of treatment. All experiments were performed 4 times; 30 cells were analyzed for each condition. Statistical analysis was done by two-way analysis of variance (ANOVA) with Tukey's test for multiple comparisons using GraphPad Prism v6 (GraphPad Software).

**Cytotoxic effect of dimAβ.** To evaluate AβO toxicity, cells were fixed and stained with NucBlue (Thermofisher Scientific) after dimAβ AβO treatment. Shape and density of nuclei were analyzed and counted: cells were considered dead, when nuclei appeared condensed and smaller, compared to viable cell nuclei. All experiments were conducted for 3 times; around 300 nuclei were analyzed for each condition. Statistical analysis was done by two-way ANOVA with Tukey's test for multiple comparisons using GraphPad Prism v6 (GraphPad Software).

**Aβ targeting to postsynaptic spines and imaging of spontaneous calcium oscillations**. To analyze Aβ binding to synapses, neurons were fixed and stained for F-actin with phalloidin as a marker of synaptic spines (Thermofisher Scientific) and a monoclonal mouse anti-Aβ (clone 4G8, Merck, #MAB1561; dilution: 1:300) antibody. The experiment was repeated independently for four times and colocalization of AβO with synapses was observed for all replicates.

To monitor spontaneous $Ca^{2+}$ oscillations, primary neurons were labeled with 2 μM Fluo-4 (Thermofisher Scientific) and 0.02% Pluronic F127 (Merck) for 20 min after 24 h of dimAβ treatment. Time-lapse movies of different fields were recorded for 1 min each (frame rate: 1 s) using a Leica DMi8 microscope (Leica) and the Leica LAS X imaging software (v3.7.3). Fluorescence intensity changes of cell bodies were quantified over time with ImageJ[74,75] and corrected for background signal. Fluorescence intensities were normalized to minimum values and peaks per minute were counted for each sample. In total, 35 cells were analyzed; statistical analysis was done by two-tailed unpaired $t$ test.

**Preparation of fluorescently labeled Aβ for cell culture experiments**. For preparation of AbberiorStar 520SXP-labeled Cys0-dimAβ, a mutant of dimAβ with an N-terminal cysteine residue was expressed as described above. For fluorophore labeling, TCEP-reduced Cys0-dimAβ lyophilisate was incubated in 200 mM HEPES pH 7.0 with a twofold molar excess of maleimide-conjugated AbberiorStar 520SXP fluorophore, which was dissolved in dimethylformamide. After 2 h of incubation, the labeled dimAβ was purified using reverse-phase HPLC. Samples were lyophilized, redissolved in HFIP, and aliquots were prepared. These aliquots were lyophilized and stored at RT for later use. Abberior STAR 520SXP-labeled AβOs were prepared from a 1:10 molar ratio of Abberior STAR 520SXP-labeled dimAβ and unlabeled dimAβ, in order to avoid that the fluorophore alters AβO properties. In all, 10 μl of 1:10 mixture of Abberior STAR 520SXP-labeled dimAβ and unlabeled dimAβ was prepared in 50 mM NaOH. Quickly, 490 μl phenol red-free Dulbecco's Modified Eagle's Medium (DMEM) supplemented with 100 U/ml penicillin–streptomycin was added, and the pH was readjusted by adding 10 μl 50 mM HCl. The final dimAβ concentration was 10 μM. The sample was quiescently incubated at 37 °C in the dark for 24 h. AβO formation was confirmed using AFM.

For Aβ42 cell culture experiments, Aβ42-HiLyte Fluor 647 (Anaspec) was dissolved in HFIP and lyophilized into smaller aliquots (30 μg). For cell culture experiments, aliquots were first dissolved in 3 μl 50 mM NaOH. In all, 544 μl phenol red-free DMEM supplemented with 100 U/ml penicillin–streptomycin was added, and the pH was recalibrated by the addition of 3 μl 50 mM HCl. To avoid exposure of the Aβ peptide to local low pH environments, the HCl was pipetted into the lid of the tube, closed, and quickly vortexed. This procedure yields a 10 μM mostly monomeric stock solution of Aβ42-HiLyte Fluor 647 suitable for cell culture experiments.

**Neuroblastoma cell culture**. SH-SY5Y cells were grown to 80% confluency in DMEM with phenol red, 10% FBS, and 100 U/ml penicillin–streptomycin in T75 flasks. Experiments were performed in Ibidi collagen IV-coated μ-Slide VI 0.4. A total of 7500 cells (250,000 cells/ml) were seeded into each channel of the slide. Cells adhered to the surface of the channels within an hour of incubation at 37 °C, 5% $CO_2$, in a humidified chamber. Subsequently, the feeding reservoirs of the channels were filled with further medium. Each day, the medium in the reservoirs was replaced with fresh medium until the cell density was satisfactory for coincubation experiments.

**Coincubation experiments and imaging**. For coincubation and imaging experiments, phenol red was removed by flushing the channels three times with phenol red-free DMEM supplemented with 100 U/ml penicillin–streptomycin. Subsequently, channels were filled with medium containing corresponding Aβ species. Cells were incubated for 24 h. Channels were flushed with fresh medium and supplemented with 50 nM Yellow HCK-123 LysoTracker. Imaging was performed either on a Leica Infinity TIRF microscope or on a confocal microscope using the Leica LAS AF software. Confocal measurements were performed using a TCS SP8 STED 3× (Leica Microsystems) equipped with an HC PL APO CS2 ×100 objective (NA 1.4) at a scan speed of 600 Hz and a line accumulation of 6. A 488 nm of a pulsed white light laser was chosen as excitation for Yellow HCK-123 LysoTracker and AbberiorSTAR520XPS. The emitted fluorescent signal was detected by counting-mode hybrid detectors in the spectral range of 500–531 nm for Yellow HCK-123 LysoTracker and 650–765 nm for AbberiorStar520SXP. Additionally, a time-gating of 0.1 ns was used to avoid laser reflection.

**Reporting summary**. Further information on research design is available in the Nature Research Reporting Summary linked to this article.

## Data availability

The cryo-EM density map of dimAβ AβOs has been deposited in the Electron Microscopy Data Bank under accession code EMD-11327. The authors declare that all the data necessary to interpret, verify, and extend the research of the article are available within the article (and Supplementary Information files). All data are available from the corresponding authors on reasonable request. Source data are provided with this paper.

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

## Acknowledgements

This project has received funding from the European Research Council under the European Union's Horizon 2020 research and innovation program, grant agreement No. 726368. We acknowledge support from the Hans und Ilse Breuer-Stiftung, the Else-Kröner-Fresenius Stiftung, and Köln Fortune. We thank Raimond B.G. Ravelli, P.J. Peters, and C. López-Iglesias for advice and helpful discussions; H. Duimel for help with sample preparation; and the M4I Division of Nanoscopy of Maastricht University for microscope access and support. We acknowledge the Center of Advanced Imaging (CAI) at the Heinrich Heine University Düsseldorf for providing access to the TCS SP8 STED 3× and support during image acquisition. WT mice were provided by CMMC animal facility (Cologne, Germany) and CECAD in vivo research facility (Cologne, Germany); live-cell imaging was conducted at the CMMC imaging facility. We thank Jennifer Klimek for excellent technical support.

## Author contributions

M.P.S., F.H., S.B., G.F.S., H.Z., and W.H. designed the experiments. M.P.S., F.H., S.B., M.Z., S.H., G.F.S., H.Z., and W.H. performed the experiments and analyzed the data. M.P.S., F.H., S.B., M.Z., G.F.S., H.Z., and W.H. wrote the manuscript. All authors commented on the manuscript.

## Funding

## Competing interests

The authors declare no competing interests.

**Additional information**

