## [Peer Review File · Nature Communications]

REVIEWER COMMENTS

Reviewer #1 (Remarks to the Author):

Schützmann et al reports a study on the pH dependent biophysical properties and the biological consequences of Amyloid beta oligomer formation. The authors investigate the formation of oligomeric species at low pH values mimicking the low pH environments of lysosomes and endosomes, and compare the behaviours of Abeta aggregation and the resulting aggregates at neutral pH. I think the work is highly relevant and the experimental data seems of high quality. Overall, I think this manuscript is a valuable contribution to the field as it may add information on the elusive oligomeric species in the amyloid field.

Below are my comments (on the biophysics and structural biology aspects) which I hope the authors will find useful in revising their manuscript.

– The authors did a good job explaining the type of oligomeric species they are investigating and how these samples are made. As they also eluded to in the introductions, Abeta is capable of forming a variety of oligomeric species, for example oligomeric species can also form transiently on-pathway, through secondary nucleation or through shedding by fibril fragmentation etc., but these species are not represented in Figure 1. Therefore, the introduction should be made clearer to better take into account the possible types of oligomeric species reported previously in the literature in order to put the investigated species in this manuscript in better context.

– The authors show convincingly that Abeta assembly is multiphasic at neutral pH and the AbetaO species dominate at lower pHs using ThT assay. However, I believe the assumptions made here is that the type of oligomeric species formed at different pHs are the same? I understand that the species will have to be different in some way, and it is not easy to show beyond doubt that the type of the AbetaOs is comparable under the different conditions. However, what are the evidence in support of this assumption?

– On similar track, what are the evidence to support that the AbetaOs generated by dimAbeta is of the same type as by monomeric Abeta42? I do think the dimAbeta approach is an elegant way to trap relevant oligomeric species of Abeta40. However, the construct is based on Abeta40 and there are many reports in the literature suggesting Abeta40 and Abeta42 aggregates can be very different entities so I do worry that dimAbeta species will be different to Abeta42 species. Again, the question is whether the AbetaOs made from dimAbeta, Abeta40 and Abeta42 are all of similar type in terms of their assembly and their biological effects under the conditions used.

– The authors report interesting AbetaO structures by cryo-EM and by AFM. If it is possible to reconstruct the shape of the AbetaOs by cryo-EM then it suggests the species are fairly common in the samples. Would the authors be able to report a relative abundance in their samples judging from their cryo-EM or AFM images?

– The authors use AFM to characterise many of their samples qualitatively. However, the images and the particles seen in the images were not quantitatively assessed in terms of their dimensions. Here, the height distribution of the particles could be assessed because comparing the dimensions of the cryo-EM reconstructions, the distributions from images in Figure 1 and later samples could present evidence on the level of structural similarity of the dimAbeta particles compared to the AbetaOs seen in the later samples.

– The apparent cluster release upon pH jump reported in Figure 7 is very interesting. How do their dimensions compare to the particles seen in the cryo-EM and AFM images in Figure 1? Also, the colour scale bar seems inconsistent as the large aggregate height of the right most panel (yellow) is 10 nm while the same area in the wide view is cyan, which is around 35 nm according to the middle colour scale bar. Height distribution quantifications and scan line sections would help in comparing and visualising the similarities and the differences in this case.

Reviewer #2 (Remarks to the Author):

Oligomeric intermediates of amyloid-beta (Abeta) peptide aggregation are believed to be crucial species in the neurotoxicity of Abeta and are believed to be the main synaptotoxic species in Alzheimer's disease. However, their formation remains somewhat elusive both structurally and kinetically, as most protocols for Abeta oligomer preparation operate at high micromolar concentrations, whereas Abeta concentrations in vivo are believed to be nanomolar and less. Here, Schutzmann et al. show that Abeta oligomers formation is vastly accelerated at pH below 7, which can be found in the endosomal / lysosomal environment. They utilize covalently linked Abeta dimers to form dodecameric Abeta assemblies, which they image by cryo EM. The reconstructed structure reveals an uneven morphology reminiscent of a potato-shaped bowl. These oligomers bind to dendritic spines, alter dendritic sprouting of tau and alter Ca-signalling, suggesting that the Abeta species may be similar to the oligomeric Abeta responsible for synaptotoxicity.

1) This study presents an intriguing first structural glimpse at oligomeric forms of Abeta and suggests a mechanism for their formation in vivo. Notably, the reconstructed structure is quite different from previously proposed beta-barrel morphologies. The uneven morphology raises the question, however, how homogeneous the oligomers are and how representative the reconstructed model is for the overall oligomer population. The population of aggregates should be measured by MALS / SEC or AUC to address these concerns. Also, do the authors observe similar structures in their Abeta42 experiments? The AFM images seem to suggest the formation of protofibrils rather than distinct small oligomers.

2) The authors argue that the oligomers formed from covalently linked Abeta dimers model the toxic Abeta oligomers in AD. They present dendritic binding data, tau missorting and Ca-oscillation data, which convincingly support the idea that di-Abeta alters these parameters when compared to Abeta 40 and buffer controls. However, the authors should include Abeta42 data in these experiments (even if this has previously been done by others) so that the reader can compare di-Abeta and Abeta42 data under the same conditions.

3) In Fig 5, dimeric fluorescent Abeta seems to form distinct aggregates in the lysosomal compartment, which are absent from the Abeta monomer control experiment. However, both co-localize with lysosomal tracker dye. Does this mean that the dimer-Abeta forms aggregates faster than the monomer in the lysosomal environment? Or is this merely an effect of the higher di-Abeta peptide concentration? The authors should add a control experiment with di-abeta at the same concentration as monomeric Abeta42 (0.1 uM) to address this question.

4) It is notable that all cell-based assays are done at 0.5 uM, which interestingly is below the COC given by the authors in vitro, but is still well above physiologic Abeta levels. FRET and single molecule assays have previously shown rapid formation of ThT negative Abeta oligomers at concentrations well below those used for ADDL preparation (see Frieden / Maiti labs). The COC for oligomer formation presented here is still micromolar and thus orders of magnitude above peptide concentrations found in vivo. Will Abeta oligomer form at physiologic concentrations at low pH? This would greatly strengthen the authors' argument for the disease relevance of their system.

5) Relatedly, the authors find that Abeta oligomerization is highly accelerated (8000 x) at low endosomal pH, when compared to pH 7.2. While that is certainly impressive, it raises the question whether the critical concentration for oligomerization is similarly lowered under these conditions, which would push the aggregation threshold into the physiologic concentration range. I would much appreciate a more complete picture of the kinetic / energetic landscape for oligomer formation.

6) I am somewhat concerned by the preparation protocol for oligomer formation from Abeta42 monomers at pH 4.5. The peptide is dissolved at pH 10.9 and then the pH is adjusted to 4.5, which takes the peptide across the isoelectric point at ~pH 5.5. This risks that the peptide crashes out of solution, initiating rapid, but poorly controlled aggregation. How did the authors control for this possibility?

Minor issues:

Scale bars in Fig 3a are barely visible

Are kinetic traces reported in Fig 1b, Fig 6a-g, Fig 7 a,b, and Fig S4 averages? What was the SD in these experiments?

Reviewer #3 (Remarks to the Author):

Strengths that make this a good paper.

- This is a well-written interesting manuscript that presents useful information concerning amyloid beta oligomers (A β O). These molecules are toxins that form from normal metabolites in the brain and they are associated with neural damage leading to Alzheimer's disease. The abstract is clear, the scholarly introduction provides helpful context, and the last figure presents a graphic model that nicely frames the cell biological perspective of the manuscript.

- A novel synthetic oligomer preparation is shown to meet benchmark criteria for use as a molecular model. These "dimA β oligomers" bind predictably to synaptic spines, induce somato-dendritic Tau missorting, and reduce oscillations in cellular Ca⁺⁺ levels, a measure of neural activity. As expected, there is no indication of induced cell death. Novel cryoEM imaging gives interesting and detailed information on the structural morphology of oligomers, revealing the bowl-shaped morphology of the smallest DimA β oligomers, which are calculated to be dodecamers. A valuable feature of the preparation is its relative stability, which also is established.

- Experiments additionally advance the concept that oligomerization is promoted in the acidic environment of endosomes and lysosomes. Results show oligomers accumulate in the lysosomes of cultured cells, while kinetic analyses using ThioT readouts are used to propose that acidic pH, as found in endosomes and lysosomes, greatly accelerates oligomerization. There is some uncertainty, however, regarding reliance on ThioT as an oligomerization assay (below).

- There is good rigor and reproducibility. Attention to details, such as the impact of agitation on assembly, underscores the credibility of the work. The paper is useful for reminding the field of issues that need continued attention and investigation. Overall, the results substantiate the value of a novel preparation for investigating mechanisms of oligomer toxicity in neuronal cultures, helping fill the need for a preparation that is stable, standardized and well-characterized.

Dealing with the following points would increase the strength of this contribution.

- Regarding the new oligomer preparation

- The text needs a minor adjustment as it incorrectly states that A β O cannot be made at low A β concentrations. There are examples in the literature to the contrary (e.g., Chang et al J Mol Neuro 2003 showed oligomerization at 10 nM A β 42; Cline et al J Neurochem 2019 showed oligomerization at 30 nM). Sensitive immunoassays are needed to detect A β O that form at these low nanomolar concentrations. As discussed further below, it would be of value to substantiate the analytics of the current paper by including such immunoassays.

- Given reports of low-dose A β preparations, the text should delete statements that the dimA β A β O provide a more physiological preparation because they form at lower doses (this is a point that really isn't the strength of the paper anyway). The text could be emended to state that in typical oligomer preps "high A β concentrations are often used" ... i.e., avoid stating that they are "required." With respect to pathophysiological relevance, it would be good to cite evidence that brain-derived A β O include A β 40 proteoforms.

- It would be helpful to state clearly which protocol for making dimA β oligomers is recommended. Are they to be made at endosomal/lysosome pH? Are they equally active if made at endosomal/lysosomal pH and then brought to 7.4? Related to this point, the authors should comment on why non-acidic pH and high concentrations were used to generate the oligomers used in Figure 1 and others; this protocol seems to be off-message with respect to the hypothesized role of endosomes/lysosomes in oligomerization.

- Endosomal/lysosomal mechanism of oligomerization

- The text states that monomeric fluorescent A β 42 traffics to and accumulates in lysosomes. There should be a mention of the evidence that this peptide stays monomeric in the culture medium and

within the cell. Have Western blots been carried out to verify this, or immunocytochemistry with oligomer-selective antibodies? Rather than a single sampling at 24 hours, a time-course to monitor uptake would be strengthening. Images with greater resolution would be appropriate. It also would make sense to do these experiments with primary neuron cultures, to compare with the other cell biology experiments presented; alternatively, it could be explained why the preferred experimental approach is to use the human neuroblastoma, a transformed PNS cell type.

- The graphic in Figure 8 is very nice, and it presents an opportunity to discuss A β concentrations in endosomes. It would be interesting to see in print a calculation based on the estimated volume of an endosome: How many molecules of A β are necessary, e.g., to generate a concentration of 1 μ M?
- Reliance on the ThioT fluorescence assay.
- The use of ThioT for monitoring oligomerization needs to be fully discussed, as it is not widely used for this purpose. The literature indicates that ThioT and other thioflavins are useful probes for fibrillar A β but poor probes for A β O $_s$. The authors should substantiate their use of ThioT by citing multiple papers in which it has been used as a sensitive assay for A β O $_s$.
- Sensitivity of the ThioT assay as well as its specificity is relevant. The current results using ThioT indicate A β must exceed 1.5 μ M for oligomerization. Antibody-based assays, though, indicate A β in nanomolar doses is sufficient for rapid oligomerization.
- The 30 minute lag in ThioT signal for A β 42 oligomerization at pH 7.2 should be discussed relative to the rapid oligomerization kinetics in the literature using, e.g., dot immunoblots employing A β O $_s$ -selective antibodies.
- Whether ThioT fluorescence is pH-sensitive and might affect interpretation of oligomerization experiments at different pH levels might also be considered. How ThioT could be used to quantify oligomer levels also might be discussed.
- With respect to experiments addressing the effects of pH, it would be a valuable for the authors to substantiate their conclusions by using A β O $_s$ -selective antibodies. Oligomer-selective antibodies are useful for sensitive detection of low levels A β O $_s$ (dot immunoblots) and for structural metrics (Western blots). Immunoassays also can be used for cell biology experiments (binding to spines; co-localization with LysoTracker). [Alternatively, a generic antibody such as 6E10 could be used to detect, or rule out, the dose-dependent presence of >50 kDa species. Experimentally, incubate monomers at various doses, ultracentrifuge with 50 kDa filter, collect aqueous retentates, assess for oligomers using dot immunoblots. NB - aqueous-stable synthetic oligomers of >50 kDa typically break down to trimers and tetramers in SDS.]
- Overall, it is strongly recommend that the conclusions regarding dose and kinetics that rely on ThioT really should be bolstered via use of immunoassays.
- Characterization by AFM
- AFM images of A β 42 oligomers made at pH 7.4 should be included. These oligomers represent the benchmark for the field. This will permit comparison with the molecular morphology of the novel dimA β O $_s$ shown in Figure 1 (which also were made at pH 7.4).
- Figure 1 shows an absence of fibrils in early stages of assembly (Fig 1c, f), which gives nice support to the contention that ThioT fluorescence can detect oligomerization (i.e., not just fibrillization, as some might think). Given the high quality AFM images in Figure 1, images of comparable quality should be shown in Figures 6 and 7, or at least a comment should be made as to why Figures 6 and 7 images are less elegant and show different morphology (Figures 6 and 7 show beads-on-a-chain structures and aggregates, but Figure 1 shows individual oligomers). The quality of images in Figures 6 and 7 make it difficult to discern the differences in morphology between putative oligomers and fibrils. It would be useful to quantify the average dimensions of the structures, but it is difficult given the imprecise nature of the images. Also, the text should explain the reasons for stating that the large aggregates are clumps of smaller A β O $_s$, as opposed, e.g., to clumps of small fibrils (Figure 6i and Figure 7c).
- Here's a devil's advocate hypothesis that the authors should argue against: In Figure 7, what if the single phase ThioT kinetics seen in low pH were due to very rapid fibrillization? Such a possibility cannot be ruled out by the current AFM data and is consistent with published studies that ThioT is a good marker for fibril formation.
- Miscellaneous
- Is it accurate to use the word "enable" in the title? It is known that A β O $_s$ can in fact be made at pH 7.4 (from the literature). Perhaps use the word "accelerates?"

- A β O_s and protofibrils are thought to be products of separate assembly pathways, as the text points out. Line 22 seems to imply A β O_s and protofibrils are virtually the same.
- The Abstract could clarify that the current experiments focus on use and properties of dimA β oligomers.
- Line 89 inadvertently refers to Figure 1 b,c.
- In Figure 2 - is NucBlue specific for neurons? Might not the dense nuclei be in microglia?
- Is it appropriate to say that "dimA β A β O_s reproduced observations made for A β O_s from A β 40?" (Line 179) Results in the manuscript indicate minimal impact for A β 40. It would be good to clarify this a bit.
- Results in Figure 5 are persuasive that fluorescent A β 42 and dimA β O_s each co-localize with Lysotracker. The labeling of the figure, though, could be more in parallel, to save the reader time in figuring out the conditions and results.

Sincerely,
Bill Klein

We thank the three reviewers for their positive assessments and for the thorough review, which helped us to improve the manuscript. This is our point-by-point response:

Reviewer #1 (Remarks to the Author):

Schützmann et al reports a study on the pH dependent biophysical properties and the biological consequences of Amyloid beta oligomer formation. The authors investigate the formation of oligomeric species at low pH values mimicking the low pH environments of lysosomes and endosomes, and compare the behaviours of Abeta aggregation and the resulting aggregates at neutral pH. I think the work is highly relevant and the experimental data seems of high quality. Overall, I think this manuscript is a valuable contribution to the field as it may add information on the elusive oligomeric species in the amyloid field.

Below are my comments (on the biophysics and structural biology aspects) which I hope the authors will find useful in revising their manuscript.

We thank the reviewer for the critical reading and helpful comments on the manuscript.

– The authors did a good job explaining the type of oligomeric species they are investigating and how these samples are made. As they also eluded to in the introductions, Abeta is capable of forming a variety of oligomeric species, for example oligomeric species can also form transiently on-pathway, through secondary nucleation or through shedding by fibril fragmentation etc., but these species are not represented in Figure 1. Therefore, the introduction should be made clearer to better take into account the possible types of oligomeric species reported previously in the literature in order to put the investigated species in this manuscript in better context.

To clarify early on that we are studying a particular type of A β oligomers, we have now added this sentence to the first paragraph of the Introduction: “We note that in this work the term A β O refers exclusively to these off-pathway oligomers and does not include other oligomeric A β species, such as those transiently formed on the pathway to amyloid fibrils, through secondary nucleation, or through shedding by fibril fragmentation (ref. 15).”

– The authors show convincingly that Abeta assembly is multiphasic at neutral pH and the AbetaO species dominate at lower pHs using ThT assay. However, I believe the assumptions made here is that the type of oligomeric species formed at different pHs are the same? I understand that the species will have to be different in some way, and it is not easy to show beyond doubt that the type of the AbetaOs is comparable under the different conditions. However, what are the evidence in support of this assumption?

We observe that the characteristics of the formation kinetics of the A β Os, including the concentration dependence, are very similar across the pH range investigated, and that the acceleration of A β O formation is a continuous process between pH 7.6 and pH 4.8 (Fig. 6a-g). This suggests that the fundamental mechanism of A β O formation is unaffected by pH reduction and that the A β Os formed at different pH values are closely related. While A β O formation is continuously accelerated upon pH reduction, we observe a sudden change in morphology by AFM around pH 6.0: Much larger A β O particles are formed below pH 6.0 (to improve clarity of illustration of this observation we added the height analysis in Fig. 6p). We interpret these large particles as clusters of smaller A β Os. This interpretation is supported by the observation that low pH A β O particles can release small A β Os at neutral pH (Fig. 8a-c). However, there may be differences between atomic-level structures and between intermolecular interactions in A β Os formed at different pH. We added

a sentence to the Discussion section: “Nevertheless, there may be differences between atomic-level structures and between intermolecular interactions in A β O_s formed at different pH, just as atomic-level structures and protofilament interfaces of amyloid fibril polymorphs can differ significantly.”

– On similar track, what are the evidence to support that the AbetaOs generated by dimAbeta is of the same type as by monomeric Abeta42? I do think the dimAbeta approach is an elegant way to trap relevant oligomeric species of Abeta40. However, the construct is based on Abeta40 and there are many reports in the literature suggesting Abeta40 and Abeta42 aggregates can be very different entities so I do worry that dimAbeta species will be different to Abeta42 species. Again, the question is whether the AbetaOs made from dimAbeta, Abeta40 and Abeta42 are all of similar type in terms of their assembly and their biological effects under the conditions used.

We completely agree that A β 40 and A β 42 aggregates can be very different. However, in terms of protofibrillar A β O_s, already the very initial works recognized a high similarity of A β 40 and A β 42 species with regard to their appearance in electron microscopy (Walsh et al., JBC, 1997, ref. 10). We showed recently that also the formation kinetics of dimA β , A β 40 and A β 42 A β O_s are similar, suggesting a similar mechanism of formation (new ref. 65 [Hasecke et al., Angew. Chem. Int. Ed., 2021]). We have now added AFM data of A β 42 A β O_s formed at neutral pH as new Fig. 8d,e, to show the similarity to the dimA β A β O_s in Fig. 1 and Fig. 6. The height of the A β 42 A β O_s in these AFM images falls in the same 3-5 nm range (Fig. 8e) as that of the dimA β A β O_s (Fig. 6p). The structural similarity may explain why dimA β A β O_s have the same effects in cell culture as reported before for A β 42 A β O_s, i.e., binding to dendritic spines, potent induction of Tau missorting, and impaired neuronal activity. We refer to the new A β 42 A β O AFM data in the first paragraph of the Results section (“AFM data of A β O_s formed from A β 42 is given in Fig. 8d,e”) and later when we compare their heights to those released from A β 42 A β O clusters formed at acidic pH and transferred to neutral pH (“The height of the cluster-released A β 42 A β O_s was 3.5-4.5 nm as measured by AFM in the dried state (Fig. 8b, c), identical to that of A β 42 A β O_s (Fig. 8d,e) and dimA β A β O_s (Fig. 6p) that were directly formed at neutral pH (Fig. 8d,e).”)

– The authors report interesting AbetaO structures by cryo-EM and by AFM. If it is possible to reconstruct the shape of the AbetaOs by cryo-EM then it suggests the species are fairly common in the samples. Would the authors be able to report a relative abundance in their samples judging from their cryo-EM or AFM images?

We estimated their relative abundance from the cryo-EM images and added the data as Fig. S1C. We mention this data now also in the Results section: “The fraction of small A β O_s was 72 \pm 12% in terms of particle number, but only ~2-3% in terms of the number of A β molecules within A β O_s (Supplementary Fig. 1c).”

– The authors use AFM to characterise many of their samples qualitatively. However, the images and the particles seen in the images were not quantitatively assessed in terms of their dimensions. Here, the height distribution of the particles could be assessed because comparing the dimensions of the cryo-EM reconstructions, the distributions from images in Figure 1 and later samples could present evidence on the level of structural similarity of the dimAbeta particles compared to the AbetaOs seen in the later samples.

We have added the height analysis as new Figs. 6p, 7f-h, 8c,e. This is a very useful addition, as it clearly illustrates i) the similarity of A β O_s formed from dimA β or A β 42, ii) the transition of A β O_s towards large clusters upon pH reduction, iii) the transition from amyloid fibrils to A β O clusters upon

an increase in A β 42 concentration at acidic pH, iv) the similarity of A β 42 A β O clusters either formed *de novo* at neutral pH or dissociated from A β O clusters that were formed at acidic pH. We refer to the height analysis at a few points in the Results section:

“In AFM, these A β O clusters have average heights of \sim 100 nm, compared to heights of \sim 4 nm observed for A β O clusters formed between pH 6.0 and 7.2 (Fig. 6p).”

“The absence of a lag-free oligomerization phase is in agreement with the observation that the COC of A β 42 in in vitro assay at neutral pH is above 10 μ M (ref. 65). Consequently, the aggregation product under this condition are amyloid fibrils (Fig. 7c,f). In contrast, at pH 4.5 lag-free aggregation occurred at a concentration of 5.4 μ M and above (Fig. 7b). The change from lag-containing to lag-free conditions at pH 4.5 was accompanied by a switch in aggregate morphology from amyloid fibril networks to large A β O clusters identical to those observed for dimA β at endo-lysosomal pH (Fig. 7d,e,g,h).”

“The height of the cluster-released A β 42 A β O clusters was 3.5-4.5 nm as measured by AFM in the dried state (Fig. 8b, c), identical to that of A β 42 A β O clusters (Fig. 8d,e) and dimA β A β O clusters (Fig. 6p) that were directly formed at neutral pH (Fig. 8d,e).”

– The apparent cluster release upon pH jump reported in Figure 7 is very interesting. How do their dimensions compare to the particles seen in the cryo-EM and AFM images in Figure 1? Also, the colour scale bar seems inconsistent as the large aggregate height of the right most panel (yellow) is 10 nm while the same area in the wide view is cyan, which is around 35 nm according to the middle colour scale bar. Height distribution quantifications and scan line sections would help in comparing and visualising the similarities and the differences in this case.

The reviewer certainly refers to Figure 8 here. The apparent inconsistency in color coding stems from the fact, that all objects $>$ 10 nm in height are displayed in yellow in the rightmost panels. We needed to zoom into the height region 0-10 nm to visualize the heights of released A β O clusters. We have now added scan lines in the new panels Fig. 8b,c to facilitate the comparison between these A β 42 A β O clusters released from clusters formed at acidic pH and dimA β A β O clusters formed at neutral pH. We have added a sentence to the Results section: “The height of the cluster-released A β 42 A β O clusters was 3.5-4.5 nm as measured by AFM in the dried state (Fig. 8b, c), identical to that of A β 42 A β O clusters (Fig. 8d,e) and dimA β A β O clusters (Fig. 6p) that were directly formed at neutral pH (Fig. 8d,e).”

Reviewer #2 (Remarks to the Author):

Oligomeric intermediates of amyloid-beta (A β) peptide aggregation are believed to be crucial species in the neurotoxicity of A β and are believed to be the main synaptotoxic species in Alzheimer's disease. However, their formation remains somewhat elusive both structurally and kinetically, as most protocols for A β oligomer preparation operate at high micromolar concentrations, whereas A β concentrations in vivo are believed to be nanomolar and less. Here, Schutzmann et al. show that A β oligomers formation is vastly accelerated at pH below 7, which can be found in the endosomal / lysosomal environment. They utilize covalently linked A β dimers to form dodecameric A β assemblies, which they image by cryo EM. The reconstructed structure reveals an uneven morphology reminiscent of a potato-shaped bowl. These oligomers bind to dendritic spines, alter dendritic sprouting of tau and alter Ca-signalling, suggesting that the A β species may be similar to the oligomeric A β responsible for synaptotoxicity.

We thank the reviewer for the critical reading and helpful comments on the manuscript.

1) This study presents an intriguing first structural glimpse at oligomeric forms of Abeta and suggests a mechanism for their formation in vivo. Notably, the reconstructed structure is quite different from previously proposed beta-barrel morphologies. The uneven morphology raises the question, however, how homogeneous the oligomers are and how representative the reconstructed model is for the overall oligomer population. The population of aggregates should be measured by MALS / SEC or AUC to address these concerns.

We fully agree that A β O are structurally heterogeneous. This heterogeneity has already been noted in the early works that showed that light scattering does not detect a single species but a broad distribution of species (Walsh et al., JBC 1997). It has also been reported that their average size increases with time (Walsh et al., JBC 1999), in line with our observations for dimA β A β O (Hasecke et al., Chem Sci 2018, Fig. 1B). We nevertheless think that it is interesting to study the structure of the small, spherical A β O, which may be biologically relevant A β O that may furthermore, upon lateral association, represent substructures of the curvilinear protofibrils. However, there is very likely considerable conformational heterogeneity also among the small, spherical A β O. This may be a reason for the limited resolution of the obtained 3D density reconstruction. A deeper analysis of the homogeneity of the small, spherical A β O is hampered by their relatively low population: They account for ~2-3% of A β molecules within A β O (new Supplementary Fig. 1c).

We have extended the Results section to clarify this point: “Structure determination is hampered by the size and shape heterogeneity of A β O (ref. 7,9,10), which is moreover evolving with time, as observed for A β O formed from A β (ref. 9) as well as dimA β (ref. 11). As larger A β O seem to be assemblies of small spherical structures, our analysis focused on the small A β O observed in the micrographs (Fig. 1d,e, Supplementary Figs. 1-3). The fraction of small A β O was 72 \pm 12% in terms of particle number, but only ~2-3% in terms of the number of A β molecules within A β O (Supplementary Fig. 1c). The relation between the small and the elongated curvilinear A β O cannot be inferred from the micrographs. Nevertheless, structure elucidation of the small A β O could provide insight into a biologically relevant A β O substructure that may furthermore laterally associate and convert into protofibrillar A β O (ref. 54).”

Also, do the authors observe similar structures in their Abeta42 experiments? The AFM images seem to suggest the formation of protofibrils rather than distinct small oligomers.

We have added AFM data of A β 42 A β O (Fig. 8d,e), which is very similar to that of dimA β A β O (Fig. 1, Fig. 6; please see also answers to Reviewer #1). We agree that it will be very interesting to study in detail the A β 42 A β O species after dissociation at neutral pH from clusters formed at acidic pH. We plan to investigate this further, but this is challenging as the A β O clusters are quite persistent, and will require more time.

2) The authors argue that the oligomers formed from covalently linked Abeta dimers model the toxic Abeta oligomers in AD. They present dendritic binding data, tau missorting and Ca-oscillation data, which convincingly support the idea that di-Abeta alters these parameters when compared to Abeta 40 and buffer controls. However, the authors should include Abeta42 data in these experiments

(even if this has previously been done by others) so that the reader can compare di-Abeta and Abeta42 data under the same conditions.

The same experiments were performed and published for A β 42 oligomers by co-author Hans Zempel (ref. 38, 39, 61). In these experiments, oligomers were prepared at the required high A β 42 concentrations and then applied to the cells at the same concentrations as done here for dimA β . As these results were already published by someone from our team using the same methods, we did not perform these experiments again.

3) In Fig 5, dimeric fluorescent Abeta seems to form distinct aggregates in the lysosomal compartment, which are absent from the Abeta monomer control experiment. However, both co-localize with lysosomal tracker dye. Does this mean that the dimer-Abeta forms aggregates faster than the monomer in the lysosomal environment? Or is this merely an effect of the higher di-Abeta peptide concentration? The authors should add a control experiment with di-abeta at the same concentration as monomeric Abeta42 (0.1 μ M) to address this question.

The apparent differences between A β 42 and dimA β in these images are largely due to the different microscopes that were used to acquire the images. DimA β images were acquired using a confocal microscope (not present in our institute) and A β 42 images were acquired using an epifluorescence microscope (present in our institute) due to COVID-19 restrictions. Consequently, vesicular compartments appear clearly defined in dimA β images but an increased background fluorescence is evident in A β 42 images. We have now acquired somewhat better quality images of the A β 42 monomer samples. Unfortunately, we were still dependent on the epifluorescence microscope, hence some background fluorescence still remains.

The different concentrations applied in the original experiments stemmed from the fact that we dilute the fluorophore during A β O preparation in order to avoid that the fluorophore alters A β O properties. We added to the Materials and Methods section: "Abberior STAR 520SXP-labeled A β O were prepared from a 1:10 molar ratio of Abberior STAR 520SXP-labeled dimA β and unlabeled dimA β , in order to avoid that the fluorophore alters A β O properties." To detect sufficient signal in microscopy, we therefore applied a higher concentration of the dimA β A β O, resulting in the same fluorophore concentration as in the A β 42 experiment (0,1 μ M). We have now repeated the A β 42 experiment but in the presence of 1 μ M unlabeled A β 42 monomer in addition to the 0.1 μ M labeled A β 42 for the sake of comparability. The revised Figure 5 shows the results of these experiments.

We also note that this experiment merely shows that both A β monomers and A β O can be taken up by the cell and accumulate in the endo-lysosomal compartment but it is not able to resolve the assembly state of A β within endo-lysosomes. We have extended the Results section to clarify this point: "In a second attempt, SH-SY5Y cells were treated with 1.1 μ M Abberior Star 520SXP-labelled dimA β A β O, formed from a mixture of 91% unlabeled and 9% fluorophore-labeled dimA β (i.e., same final concentrations of unlabeled and fluorophore-labeled A β as in the A β 42 experiment above). This experiment revealed a similar colocalization in acidic vesicles as for A β 42 (Fig. 5). This confirms that both A β monomers and A β O are readily taken up by neuron-like cells and accumulate in the endo-lysosomal system. Our results, however, do not reveal the assembly state of A β , and it is possible that the applied A β species undergo structural alterations upon cell entry and accumulation in endo-lysosomes, such as higher-order assembly as described below."

4) It is notable that all cell-based assays are done at 0.5 μ M, which interestingly is below the COC given by the authors in vitro, but is still well above physiologic Abeta levels. FRET and single molecule

assays have previously shown rapid formation of ThT negative Abeta oligomers at concentrations well below those used for ADDL preparation (see Frieden / Maiti labs). The COC for oligomer formation presented here is still micromolar and thus orders of magnitude above peptide concentrations found in vivo. Will Abeta oligomer form at physiologic concentrations at low pH? This would greatly strengthen the authors' argument for the disease relevance of their system.

Yes, we argue in this paper that A β O form at physiological concentrations at low pH. In our in vitro experiments, we find that the COC of A β 42 is approximately 5 μ M at pH 4.5. While this is orders of magnitudes higher than the A β level in brain interstitial fluid, which is usually referred to as 'physiological A β level', it may well be below the physiological concentration of A β in the endo-lysosomal system. Hu, Frieden, Pappu, Lee et al. (ref. 44) had a detailed look at the relevant A β concentration in the endo-lysosomal system and determined it to be well above 2.5 μ M. Due to technical limitations associated with fluorescence intensity readouts, this was a conservative lower limit estimate for several reasons, as the authors pointed out in the Discussion section of their paper, and they concluded "Therefore, it is likely that the concentrations of A β in the intracellular vesicles may, in fact, be an order of magnitude higher than what we have estimated." We have added to the Discussion section: "At the same time, the COC of A β O formation is reduced. This enables A β O formation at physiologically relevant A β concentrations, determined to be well above 2.5 μ M in endo-lysosomal vesicles (ref. 44)."

5) Relatedly, the authors find that Abeta oligomerization is highly accelerated (8000 x) at low endosomal pH, when compared to pH 7.2. While that is certainly impressive, it raises the question whether the critical concentration for oligomerization is similarly lowered under these conditions, which would push the aggregation threshold into the physiologic concentration range. I would much appreciate a more complete picture of the kinetic / energetic landscape for oligomer formation.

We had a closer look at the effect of pH reduction on the COC of both dimA β and A β 42 using the ThT assay. The dimA β data is presented as new Supplementary Fig. 6, the associated paragraph in the Results section reads "In order to test whether the acceleration of A β O formation kinetics is accompanied by thermodynamic stabilization, we evaluated the effect of pH reduction on the COC of dimA β . In the A β O formation assay at pH 7.4, the fluorescence intensity increase during the lag-free oligomerization phase scaled linearly with protein concentration at dimA β concentrations above \sim 2 μ M, whereas no lag-free oligomerization was detectable below \sim 0.5 μ M, indicative of a COC of around 1 μ M (Supplementary Fig. 6a, b). At pH 5.6, however, there is no indication of disappearance of the oligomerization phase down to a concentration of 0.4 μ M dimA β (Supplementary Fig. 6c, d). Due to the limited sensitivity of ThT at acidic pH (ref. 64) it is not possible to reliably monitor oligomerization at lower concentrations and to determine the COC at this pH. Nevertheless, the COC at pH 5.6 is clearly lower than the COC at neutral pH, indicative of thermodynamic stabilization of A β O at acidic pH."

For A β 42, we have recently determined the COC at neutral pH to fall into the range 10-30 μ M (new ref. 65 [Hasecke et al., Angew. Chem. Int. Ed., 2021]). We compare the COC at pH 4.5 with this new publication in a sentence introduced in the paragraph *A β O assembly of A β 42 is enabled under endo-lysosomal conditions* in the Results section: "The absence of a lag-free oligomerization phase is in agreement with the observation that the COC of A β 42 in in vitro assay at neutral pH is above 10 μ M (ref. 65)."

6) I am somewhat concerned by the preparation protocol for oligomer formation from Abeta42 monomers at pH 4.5. The peptide is dissolved at pH 10.9 and then the pH is adjusted to 4.5, which

takes the peptide across the isoelectric point at ~pH 5.5. This risks that the peptide crashes out of solution, initiating rapid, but poorly controlled aggregation. How did the authors control for this possibility?

The pH dependent data of dimA β assembly kinetics indicates that decreasing the pH continuously accelerates A β O formation, without any apparent change in mechanism (e.g., in the reaction order) when crossing the isoelectric point (Fig. 6). The A β 42 kinetics at pH 4.5 look similar to those of dimA β at acidic pH, and so do the resulting assemblies, which probably represent clusters of A β O_s. The pH shift experiment demonstrates that these clusters release small A β O_s upon pH neutralization. We therefore don't find evidence for a shift towards 'poorly controlled' aggregation around the pI. However, the clustering of A β O_s into large assemblies occurs around the pI. It might well be that this higher order assembly is most pronounced around the pI and driven by low net charge.

Minor issues:

Scale bars in Fig 3a are barely visible

We increased the size of the scale bar.

Are kinetic traces reported in Fig 1b, Fig 6a-g, Fig 7 a,b, and Fig S4 averages? What was the SD in these experiments?

The time traces shown were/are individual representative examples. For Fig. 1b, we have now directly added replicates. This illustrates the good reproducibility of the nucleation-free oligomerization phase, and the stochastic nature of the nucleation-dependent fibril growth phase. For Figs. 6 and 7 we have added the replicates sets as Supplementary Figs. 8 and 9. We have also added replicates to Supplementary Fig. 4.

Reviewer #3 (Remarks to the Author):

Strengths that make this a good paper.

- This is a well-written interesting manuscript that presents useful information concerning amyloid beta oligomers (A β O_s). These molecules are toxins that form from normal metabolites in the brain and they are associated with neural damage leading to Alzheimer's disease. The abstract is clear, the scholarly introduction provides helpful context, and the last figure presents a graphic model that nicely frames the cell biological perspective of the manuscript.
- A novel synthetic oligomer preparation is shown to meet benchmark criteria for use as a molecular model. These "dimA β oligomers" bind predictably to synaptic spines, induce somato-dendritic Tau missorting, and reduce oscillations in cellular Ca⁺⁺ levels, a measure of neural activity. As expected, there is no indication of induced cell death. Novel cryoEM imaging gives interesting and detailed information on the structural morphology of oligomers, revealing the bowl-shaped morphology of the smallest DimA β oligomers, which are calculated to be dodecamers. A valuable feature of the preparation is its relative stability, which also is established.
- Experiments additionally advance the concept that oligomerization is promoted in the acidic environment of endosomes and lysosomes. Results show oligomers accumulate in the lysosomes of cultured cells, while kinetic analyses using ThioT readouts are used to propose that acidic pH, as found in endosomes and lysosomes, greatly accelerates oligomerization. There is some uncertainty, however, regarding reliance on ThioT as an oligomerization assay (below).

- There is good rigor and reproducibility. Attention to details, such as the impact of agitation on assembly, underscores the credibility of the work. The paper is useful for reminding the field of issues that need continued attention and investigation. Overall, the results substantiate the value of a novel preparation for investigating mechanisms of oligomer toxicity in neuronal cultures, helping fill the need for a preparation that is stable, standardized and well-characterized. Dealing with the following points would increase the strength of this contribution.

We thank Dr Klein for the critical reading and helpful comments on the manuscript.

- Regarding the new oligomer preparation
- The text needs a minor adjustment as it incorrectly states that A β O_s cannot be made at low A β concentrations. There are examples in the literature to the contrary (e.g., Chang et al J Mol Neuro 2003 showed oligomerization at 10 nM A β 42; Cline et al J Neurochem 2019 showed oligomerization at 30 nM). Sensitive immunoassays are needed to detect A β O_s that form at these low nanomolar concentrations. As discussed further below, it would be of value to substantiate the analytics of the current paper by including such immunoassays.

We thank the reviewer for raising the point that we should put our study into the perspective of the low concentration A β O_s observed in some studies before. We agree that we are studying a particular type of A β O_s here. The high concentration dependence of their formation results in a rather well-defined critical concentration (COC), which is in the μ M range at neutral pH in *in vitro* assays. This has two consequences: Formation of these A β O_s below the COC is an extremely rare event. On the other hand, above the COC a large fraction of A β is present as A β O. We think that it is not so important to detect the very low concentrations of these A β O_s that might be present below the COC under *in vitro* conditions. It is more interesting to elucidate which *in vivo* conditions shift the COC to lower values, thereby enabling formation of large amounts of these A β O_s with neurotoxic activity. We think that our set of experiments is sufficient to describe the effect of acidic pH on the formation and COC of these A β O_s, which supports that they might form in large amounts under endo-lysosomal conditions. We therefore do not see the necessity for sensitive immunoassays. With regard to previously observed oligomers at low A β concentrations it is possible that these are related to the A β O_s studied here, with their formation possibly promoted by the respective experimental conditions (e.g., interfaces in the experimental setup). Alternatively, it is also possible that these low concentration oligomers represent distinct types of oligomers.

We have added a sentence to the first paragraph of the Introduction, also in response to Q1 of reviewer 1: “We note that in this work the term A β O refers exclusively to these off-pathway oligomers and does not include other oligomeric A β species, such as those transiently formed on the pathway to amyloid fibrils, through secondary nucleation, or through shedding by fibril fragmentation (ref. 15).”

- Given reports of low-dose A β preparations, the text should delete statements that the dimA β A β O_s provide a more physiological preparation because they form at lower doses (this is a point that really isn't the strength of the paper anyway). The text could be emended to state that in typical oligomer preps “high A β concentrations are often used” ... i.e., avoid stating that they are “required.” With respect to pathophysiological relevance, it would be good to cite evidence that brain-derived A β O_s include A β 40 proteoforms.

We have modified the statements to now include reference to the COC aspect:

“At neutral pH, high A β concentrations are required to convert a substantial fraction of the protein into A β Os.”

“There is an apparent discrepancy between the obvious pathogenic relevance of A β Os and the high μ M A β concentrations required for the conversion of a substantial fraction of the protein into A β Os at neutral pH in vitro”

We note that we also show A β O formation of A β 42. In terms of protofibrillar A β Os, already the very initial works recognized a high similarity of A β 40 and A β 42 species with regard to their appearance in electron microscopy (Walsh et al., JBC, 1997, ref. 10). We showed recently that also the formation kinetics of dimA β , A β 40 and A β 42 A β Os are similar, suggesting a similar mechanism of formation (new ref. 65 [Hasecke et al., Angew. Chem. Int. Ed., 2021]). We have now added AFM data of A β 42 A β Os formed at neutral pH as new Fig. 8d,e, to show the similarity to the dimA β A β Os in Fig. 1 and Fig. 6. The height of the A β 42 A β Os in these AFM images falls in the same 3-5 nm range (Fig. 8e) as that of the dimA β A β Os (Fig. 6p). The structural similarity may explain why dimA β A β Os have the same effects in cell culture as reported before for A β 42 A β Os, i.e., binding dendritic spines, potent induction of Tau missorting, and impaired neuronal activity.

- It would be helpful to state clearly which protocol for making dimA β oligomers is recommended. Are they to be made at endosomal/lysosome pH? Are they equally active if made at endosomal/lysosomal pH and then brought to 7.4? Related to this point, the authors should comment on why non-acidic pH and high concentrations were used to generate the oligomers used in Figure 1 and others; this protocol seems to be off-message with respect to the hypothesized role of endosomes/lysosomes in oligomerization.

As A β Os are present both at a neutral and at endo-lysosomal pH values in the physiological situation, we think it is relevant to study this whole pH range. An advantage of the dimA β A β Os produced at neutral pH is the absence of clustering that is observed for A β Os produced at low pH. A general advantage of the dimA β system is that thanks to the low COC, solutions consisting almost exclusively of A β Os, without contaminations by amyloid fibrils and containing only low levels of monomers, can be produced. For example, in the 20 μ M dimA β solutions used as stocks for the cell culture experiments A β Os are the dominant species by far. We exploit this advantage in the first part of the paper, where we compare the effects of dimA β A β Os prepared at neutral pH on neuronal cell cultures with those of previous A β 42 A β O preparations, which were also made at neutral pH. This part of the paper serves to establish the relevance of dimA β as A β O model. The subsequent part on the pH dependence indeed suggests a particular relevance of A β Os formed at low pH. We agree that it will be very interesting to study in detail the activities of A β Os after dissociation at neutral pH from clusters formed at acidic pH. The similarity in appearance under the AFM to A β Os formed at neutral pH may suggest that these two species might behave similar, but this needs to be tested. We plan to investigate this further, but this is challenging as the A β O clusters are quite persistent, and will require more time.

- Endosomal/lysosomal mechanism of oligomerization
- The text states that monomeric fluorescent A β 42 traffics to and accumulates in lysosomes. There should be a mention of the evidence that this peptide stays monomeric in the culture medium and within the cell. Have Western blots been carried out to verify this, or immunocytochemistry with oligomer-selective antibodies? Rather than a single sampling at 24 hours, a time-course to monitor uptake would be strengthening. Images with greater resolution would be appropriate. It also would make sense to do these experiments with primary neuron cultures, to compare with the other cell

biology experiments presented; alternatively, it could be explained why the preferred experimental approach is to use the human neuroblastoma, a transformed PNS cell type.

The aim of this section is to show that dimA β A β O are taken up into endo-lysosomal compartments, just like it has been described for A β 42 in several studies using neuroblastoma cells or primary neurons (ref. 41,44-46). We did not want to speculate on the assembly state of A β 42 during entry into the cells and trafficking into endo-lysosomes. We realize that the previous section title may have been misleading and have removed the mentions of assembly states. The section title now reads "A β 42 as well as dimA β accumulate within endo-lysosomal compartments". We have furthermore added the following sentence to this paragraph: "Our results, however, do not reveal the assembly state of A β , and it is possible that the applied A β species undergo structural alterations upon cell entry and accumulation in endo-lysosomes, such as higher-order assembly as described below."

- The graphic in Figure 8 is very nice, and it presents an opportunity to discuss A β concentrations in endosomes. It would be interesting to see in print a calculation based on the estimated volume of an endosome: How many molecules of A β are necessary, e.g., to generate a concentration of 1 μ M?

We have added this calculation to the legend of the figure (now Fig. 9): "Using a conservative estimate of the endo-lysosomal A β concentration of 2.5 μ M (ref. 44) and assuming an endosome volume of 0.3 μ m³, there are on average 450 A β molecules in an endosome."

- Reliance on the ThioT fluorescence assay.
- The use of ThioT for monitoring oligomerization needs to be fully discussed, as it is not widely used for this purpose. The literature indicates that ThioT and other thioflavins are useful probes for fibrillar A β but poor probes for A β O. The authors should substantiate their use of ThioT by citing multiple papers in which it has been used as a sensitive assay for A β O.

There is extensive literature now on the detection of the amyloid oligomers of the globular oligomer/curvilinear fibril type using ThT. For example, our collaboration partner Martin Muschol has investigated this in detail for different amyloid systems, including A β . In a recent paper, the Muschol lab compared ThT with other dyes regarding their sensitivities for oligomers vs. amyloid fibrils (Barton et al., *Biomolecules*, 2019, doi: 10.3390/biom9100539). We also believe that the present manuscript and our previous papers on dimA β and A β assembly (ref. 11 [Hasecke et al., *Chem. Sci.*, 2018]; ref. 65 [Hasecke et al., *Angew. Chem. Int. Ed.*, 2021]) provide sufficient evidence that ThT can detect A β oligomers of the globular oligomer/curvilinear fibril type.

- Sensitivity of the ThioT assay as well as its specificity is relevant. The current results using ThioT indicate A β must exceed 1.5 μ M for oligomerization. Antibody-based assays, though, indicate A β in nanomolar doses is sufficient for rapid oligomerization.

The sensitivity of ThT for A β O is sufficient to detect a true critical concentration for oligomer formation at neutral pH, as can be seen in the new Supplementary Fig. 6 (this is already evident from the data in ref. 11 [Hasecke et al., *Chem. Sci.*, 2018] and discussed in that paper). This critical concentration is inherent to the oligomerization reaction; it stems from the high reaction order of oligomer formation. It is not an apparent critical concentration due to a lack of sensitivity of the assay. For the relation of the A β O studied here to A β O observed at low A β concentrations with immunoassays, please see our answer to Q1.

- The 30 minute lag in ThioT signal for A β 42 oligomerization at pH 7.2 should be discussed relative to the rapid oligomerization kinetics in the literature using, e.g., dot immunoblots employing A β O-selective antibodies.

The A β 42 kinetics at pH 7.2 were performed at concentrations <10 μ M. In a recent paper (ref. 64 [Hasecke et al., *Angew. Chem. Int. Ed.*, 2021]) we have shown that the COC of A β 42 falls in the 10-30 μ M range under these conditions. We therefore do not expect to see A β O of the type studied in this paper under these conditions. The 30 minute lag in the ThT signal is the lag-time of nucleated fibril formation.

- Whether ThioT fluorescence is pH-sensitive and might affect interpretation of oligomerization experiments at different pH levels might also be considered. How ThioT could be used to quantify oligomer levels also might be discussed.

ThT fluorescence is pH dependent, and we mention this at the points in the manuscript where it is of relevance (ref. 64 [Hackl et al.]). The pH sensitivity does not affect the determination of kinetic constants, as this is independent of absolute fluorescence intensity values. The pH sensitivity does also not affect determination of the COC for dimA β at pH 7.2 (new Supplementary Fig. 6a, b). At a fixed pH, ThT can well be used to quantify A β O levels, as is evident from the linear correlation of ThT fluorescence with dimA β concentration above the COC (new Supplementary Fig. 6b, inset). The ThT sensitivity is limiting, however, when we try to determine the COC of dimA β at pH 5.6, since we cannot reliably detect oligomerization below 0.4 μ M dimA β at this pH. We discuss this limitation in the revised manuscript: “Due to the limited sensitivity of ThT at acidic pH (ref. 64) it is not possible to reliably monitor oligomerization at lower concentrations and to determine the COC at this pH. Nevertheless, the COC at pH 5.6 is clearly lower than the COC at neutral pH, indicative of thermodynamic stabilization of A β O at acidic pH.”

- With respect to experiments addressing the effects of pH, it would be a valuable for the authors to substantiate their conclusions by using A β O-selective antibodies. Oligomer-selective antibodies are useful for sensitive detection of low levels A β O (dot immunoblots) and for structural metrics (Western blots). Immunoassays also can be used for cell biology experiments (binding to spines; co-localization with LysoTracker). [Alternatively, a generic antibody such as 6E10 could be used to detect, or rule out, the dose-dependent presence of >50 kDa species. Experimentally, incubate monomers at various doses, ultracentrifuge with 50 kDa filter, collect aqueous retentates, assess for oligomers using dot immunoblots. NB - aqueous-stable synthetic oligomers of >50 kDa typically break down to trimers and tetramers in SDS.]

We thank the reviewer for these suggestions. As pointed out already in our answer to Q1, we are studying a particular type of A β O here, which is characterized by a COC, above which a substantial fraction of A β is present in the A β O state. We are studying the effects of these A β O prepared above the COC, and the effect of pH on the COC. Our methods are sufficiently sensitive to detect the (relatively large) amounts of A β O relevant for these tasks. We also want to point out that the characteristic formation kinetics, whose observation is the basis of our study, results in a structure specific analysis, as we analyze A β O that have formed through a common mechanism. Nevertheless, antibodies selective for these A β O will certainly be interesting for various immunoassays, and we plan to generate such antibodies in the future.

- Overall, it is strongly recommend that the conclusions regarding dose and kinetics that rely on ThioT really should be bolstered via use of immunoassays.

As argued above, we are convinced that the conclusions of our paper are sufficiently substantiated by the methods we applied. We do not see the critical question in the context of this manuscript that immunoassays would answer. However, we completely agree that it would be highly interesting for future studies to develop immunoassays that specifically detect the A β O_s investigated in the present study.

- Characterization by AFM
- AFM images of A β 42 oligomers made at pH 7.4 should be included. These oligomers represent the benchmark for the field. This will permit comparison with the molecular morphology of the novel dimA β O_s shown in Figure 1 (which also were made at pH 7.4).

We have now added AFM data of A β 42 A β O_s formed at neutral pH as new Fig. 8d,e, to show the similarity to the dimA β A β O_s in Fig. 1 and Fig. 6. The height of the A β 42 A β O_s in these AFM images falls in the same 3-5 nm range (Fig. 8e) as that of the dimA β A β O_s (Fig. 6p).

- Figure 1 shows an absence of fibrils in early stages of assembly (Fig 1c, f), which gives nice support to the contention that ThioT fluorescence can detect oligomerization (i.e., not just fibrillization, as some might think). Given the high quality AFM images in Figure 1, images of comparable quality should be shown in Figures 6 and 7, or at least a comment should be made as to why Figures 6 and 7 images are less elegant and show different morphology (Figures 6 and 7 show beads-on-a-chain structures and aggregates, but Figure 1 shows individual oligomers). The quality of images in Figures 6 and 7 make it difficult to discern the differences in morphology between putative oligomers and fibrils. It would be useful to quantify the average dimensions of the structures, but it is difficult given the imprecise nature of the images. Also, the text should explain the reasons for stating that the large aggregates are clumps of smaller A β O_s, as opposed, e.g., to clumps of small fibrils (Figure 6i and Figure 7c).

In the revised Figs. 6 and 7, we have replaced some AFM panels with images of higher quality and added a height analysis as new Figs. 6p, 7f-h, 8c,e. This is a very useful addition, as it clearly illustrates i) the similarity of A β O_s formed from dimA β or A β 42, ii) the transition of A β O_s towards large clusters upon pH reduction, iii) the transition from amyloid fibrils to A β O clusters upon an increase in A β 42 concentration at acidic pH, iv) the similarity of A β 42 A β O_s either formed *de novo* at neutral pH or dissociated from A β O clusters that were formed at acidic pH.

The explanation why we think the large aggregates are clumps of smaller A β O_s is given in the sentences "Here, we found that a reaction order of 3 applied to global fitting of the concentration-dependent data results in fits that reproduce the kinetic traces at all pH values (Fig. 6a-g). This indicates that the fundamental mechanism of A β O formation is not affected by pH reduction. [...] Below pH 6.0, A β O_s associated into large clusters, in line with a previous description of A β 40 aggregates at pH 5.8 (ref. 53). Thus, while the fundamental mechanism of A β O formation seems to be unaffected by pH reduction, there is an additional level of particle aggregation involved below pH 6.0."

- Here's a devil's advocate hypothesis that the authors should argue against: In Figure 7, what if the single phase ThioT kinetics seen in low pH were due to very rapid fibrillization? Such a possibility

cannot be ruled out by the current AFM data and is consistent with published studies that ThioT is a good marker for fibril formation.

The AFM images of the low and high concentration A β 42 assemblies at pH 4.5 are clearly different, which is now more evident thanks to the added height profiles. The low concentration assemblies are in line with amyloid fibrils, the high concentration assemblies are not. Furthermore, upon shifting the pH of the high concentration assemblies to neutral, small A β O are released (Fig. 8).

- Miscellaneous

- Is it accurate to use the word “enable” in the title? It is known that A β O can in fact be made at pH 7.4 (from the literature). Perhaps use the word “accelerates?”

As discussed above, the type of A β O that we study have a rather high COC of 10-30 μ M for A β 42 under in vitro conditions. Therefore, in vivo factors are required to enable the formation of any substantial amounts of these A β O. We can well imagine that A β O formation can also be enabled at neutral pH, e.g. by increased local concentration, specific surfaces, or other co-factors. Our study shows that two factors that might well enable A β O formation are endo-lysosomal pH and high local endo-lysosomal A β concentration. We therefore believe that the title is accurate. However, we now suggest to replace “enable” by “trigger”: “Endo-lysosomal A β concentration and pH trigger formation of A β oligomers that potentially induce Tau missorting”

- A β O and protofibrils are thought to be products of separate assembly pathways, as the text points out. Line 22 seems to imply A β O and protofibrils are virtually the same.

We have deleted the term “protofibril” from the abstract. The term is referred to in the Introduction, which should be sufficient.

- The Abstract could clarify that the current experiments focus on use and properties of dimA β oligomers.

We think the abstract gives a fair account of the main points of this study and would like to keep its current structure. First, The A β 42 data in Figs. 7 and 8 is a very crucial part of this study. Second, the abstract mentions that the dimeric construct dimA β was applied. Third, if we would mention dimA β first and A β 42 second, the reader would be misled to imagine a dimeric A β 42 construct.

- Line 89 inadvertently refers to Figure 1 b,c.

This reference was placed here on purpose. Fig. 1 b,c displays kinetics and AFM images of A β O formation, representing the basic observations made in ref. 11.

- In Figure 2 - is NucBlue specific for neurons? Might not the dense nuclei be in microglia?

We agree with reviewer 3 that NucBlue is not specific for neuronal nuclei. However, due to our preparation and cultivation protocol with specialized media for primary neurons, which involves treatment with Cytosine β -D-arabino-furanoside (AraC), an anti-metabolic drug that eliminates dividing cells (but not postmitotic neurons) from neuronal cultures, we have no or negligible amounts of glial cells in our neuronal cultures: >90% of cells are neurons, ~3% oligodendrocytes and <0.1% microglia, as assessed by immunofluorescent stainings for corresponding marker proteins, such as Tau, Olig2 and IBA1. These numbers are comparable to other published protocols, e.g.

Beaudoin, G. M. 3rd et al. (2012); Moutin E et al. (2020); Ray J. et al. (1993). In addition, glial nuclei can be easily distinguished from neuronal nuclei, as they differ significantly in size. Since we also co-stained for Tau in our experiments we can distinguish neuronal from glial cells by the levels of Tau expression, which is extremely low in glial cells (except for oligodendrocytes, but these cells can easily be distinguished due to their particular morphology).

Beaudoin GM 3rd, Lee SH, Singh D, Yuan Y, Ng YG, Reichardt LF, Arikath J. Culturing pyramidal neurons from the early postnatal mouse hippocampus and cortex. *Nat Protoc.* 2012 Sep;7(9):1741-54. doi:10.1038/nprot.2012.099.

Moutin, E., Hemonnot, A. L., Seube, V., Linck, N., Rassendren, F., Perroy, J., & Compan, V. (2020). Procedures for culturing and genetically manipulating murine hippocampal postnatal neurons. *Front Synaptic Neurosci*, 12, 19.

Ray, J., Peterson, D. A., Schinstine, M., & Gage, F. H. (1993). Proliferation, differentiation, and long-term culture of primary hippocampal neurons. *Proc Natl Acad Sci USA*, 90(8), 3602–3606.

- Is it appropriate to say that “dimA β A β O_s reproduced observations made for A β O_s from A β 40?” (Line 179) Results in the manuscript indicate minimal impact for A β 40. It would be good to clarify this a bit.

This statement refers to A β O_s prepared from A β 40 as described in the literature, not to the monomeric A β 40 control experiments in the present manuscript. The paper reference here is ref. 38 [Zempel et al., 2013, *EMBO J*], which reports similar effects of A β O_s formed from A β 40 and A β 42, and a threefold stronger effect of the 7:3 A β 40:A β 42 mixtures on tau missorting (Supplementary Results (a) and Supplementary Fig. 2e, f in that paper). We have added a “previously” to the sentence to make clearer that we are referring to the cited literature. The sentence now reads “With regard to dendritic spine binding, lack of direct cytotoxicity, potent induction of Tau missorting as well as decreased neuronal activity, dimA β A β O_s thus faithfully reproduce the observations previously made for A β O_s formed from A β 40 or A β 42, or from 7:3 A β 40:A β 42 mixtures regarded as particularly toxic (ref. 38).”

- Results in Figure 5 are persuasive that fluorescent A β 42 and dimA β O_s each co-localize with Lysotracker. The labeling of the figure, though, could be more in parallel, to save the reader time in figuring out the conditions and results.

We have modified Fig. 5 accordingly.

REVIEWER COMMENTS

Reviewer #1 (Remarks to the Author):

The authors have addressed all of my previous comments. In particular, I think the height analysis of the structures observed in the AFM images has improved the overall interpretation and presentation of the data. I have only two minor comments that I suggest the authors consider before publication: 1. It would help the readers if the very short description of image segmentation using MorphACWE function is complemented with illustrative examples of how the boundaries levels look on a pixel level before and after segmentation, perhaps as an SI figure. 2. The population estimate for figure 1 / S1 is very helpful. However, any structural reconstruction is only meaningful if the particles are, at least to some extent, structurally homogeneous, which they are not, even according to the authors' own assessment. Therefore, I would suggest further sentences in the figure legend of Fig 1 to emphasize to the readers that the shapes in Fig 1d and e are only rough guide to size and volume of the oligomeric species in this case.

Reviewer #2 (Remarks to the Author):

The authors have addressed my technical concerns in the revised manuscript, which strengthens their conclusions and results in an impactful and important study. I support its publication in its current form.

Reviewer #3 (Remarks to the Author):

The authors did a fine job at handling a rigorous set of comments from me and the other reviewers. I am more than satisfied with the emendations, or with the explanations as to why particular emendations were not needed for this paper. As before, I believe this is an excellent study.

We thank the three reviewers for their thorough reviews and for supporting the publication of this manuscript.

Reviewer #1 (Remarks to the Author):

The authors have addressed all of my previous comments. In particular, I think the height analysis of the structures observed in the AFM images has improved the overall interpretation and presentation of the data. I have only two minor comments that I suggest the authors consider before publication:

1. It would help the readers if the very short description of image segmentation using MorphACWE function is complemented with illustrative examples of how the boundaries levels look on a pixel level before and after segmentation, perhaps as an SI figure.

We have added this as Supplementary Fig. S10, which is referenced in the Methods section.

2. The population estimate for figure 1 / S1 is very helpful. However, any structural reconstruction is only meaningful if the particles are, at least to some extent, structurally homogeneous, which they are not, even according to the authors' own assessment. Therefore, I would suggest further sentences in the figure legend of Fig 1 to emphasis to the readers that the shapes in Fig 1d and e are only rough guide to size and volume of the oligomeric species in this case.

We have added two sentences to the legend of Fig. 1: "The comparatively low resolution is due to the small size and high degree of heterogeneity of the dimA β A β O species. Consequently, only a rough estimate to size and volume can be made."